# Structural features stabilized by divalent cation coordination within hepatitis E virus ORF1 are critical for viral replication

Robert LeDesma[1], Brigitte Heller[1], Abhishek Biswas[1], Stephanie Maya[1], Stefania Gili[2], John Higgins[2], Alexander Ploss[1]*

[1]Department of Molecular Biology, Lewis Thomas Laboratory, Princeton University, Princeton, United States; [2]Department of Geosciences, Princeton University, Princeton, United States

**Abstract** Hepatitis E virus (HEV) is an RNA virus responsible for over 20 million infections annually. HEV's open reading frame (ORF)1 polyprotein is essential for genome replication, though it is unknown how the different subdomains function within a structural context. Our data show that ORF1 operates as a multifunctional protein, which is not subject to proteolytic processing. Supporting this model, scanning mutagenesis performed on the putative papain-like cysteine protease (pPCP) domain revealed six cysteines essential for viral replication. Our data are consistent with their role in divalent metal ion coordination, which governs local and interdomain interactions that are critical for the overall structure of ORF1; furthermore, the 'pPCP' domain can only rescue viral genome replication in trans when expressed in the context of the full-length ORF1 protein but not as an individual subdomain. Taken together, our work provides a comprehensive model of the structure and function of HEV ORF1.

*For correspondence:
aploss@princeton.edu

Competing interest: The authors declare that no competing interests exist.

## Editor's evaluation

Your findings that polyprotein domains are likely to have exclusively structural functions is important to the field. It is often not appreciated that a large polyprotein is not merely a linear assembly of the final digestion products and must adopt particular conformations to support the ordered cleavages that occur.

## Introduction

Hepatitis E virus (HEV) has a global disease burden of over 20 million cases per annum, leading to approximately 70,000 fatalities (*Rein et al., 2012*). This burden is especially pronounced in immunocompromised individuals and pregnant women, the latter of whom experience a close to 30% mortality rate in the third trimester (*Khuroo and Kamili, 2003*) and/or approximately 3000 stillbirths (*Rein et al., 2012*). HEV infection can be prevented with a prophylactic vaccine which is currently only licensed in China. Presently, treatment is limited to ribavirin (RBV; *Lee and Hung, 2014*) and pegylated type I interferon (IFN; *Kamar et al., 2010*). However, these therapies are plagued with considerable side effects, as RBV is teratogenic and thus cannot be administered during pregnancy, and IFN therapy can lead to transplant rejection in organ transplant recipients (*Haagsma et al., 2010*). Furthermore, HEV strains with fitness-enhancing mutations have been identified in patients showing clinical resistance to RBV treatment (*Todt et al., 2018*). One clinical case study has suggested that sofosbuvir, a drug approved for hepatitis C virus (HCV) treatment, may have a beneficial additive effect when used in combination with RBV; however, other studies have observed

no therapeutic benefit, and the use of this drug for HEV treatment remains controversial (*van der Valk et al., 2017*; *van Wezel et al., 2019*). Other clinical drugs are currently not approved for the treatment of HEV, obviating the need for direct acting antivirals and a better understanding of the viral replication cycle.

HEV is a (+) ssRNA virus in the *Orthohepevirus* genus in the *Hepeviridae* family of viruses (*Smith et al., 2014*). The genome of HEV has a 5'-methylated cap and a 3'-poly (A) tail and is composed of three partially overlapping open reading frames (ORFs). ORF1 encodes the viral replicase (*Koonin et al., 1992*), ORF2 encodes the capsid protein (*Reyes et al., 1993*; *Tam et al., 1991*), and ORF3 encodes a viroporin necessary for viral egress (*Ding et al., 2017*). While the primary functions of ORF2 and ORF3 have been characterized, much of ORF1 remains to be understood. ORF1 has been organized into seven domains based on prior bioinformatic analysis (*Figure 1A*; *Koonin et al., 1992*), only four of which have been functionally characterized in some detail: namely the methyltransferase (*Magden et al., 2001*), macrodomain (*Parvez, 2015*), helicase (*Devhare et al., 2014*; *Karpe and Lole, 2010*), and RNA-dependent RNA polymerase (RdRp; *Koonin, 1991*; *van der Heijden and Bol, 2002*). Of the remaining domains of HEV ORF1, the putative papain-like cysteine protease (pPCP) has been the subject of some debate; several groups assessing the functionality of this region have disagreed on the presence of protease activity (*Ansari et al., 2000*; *Kanade et al., 2018*; *Karpe and Lole, 2011*; *Paliwal et al., 2014*; *Parvez, 2013*; *Parvez and Khan, 2014*; *Perttilä et al., 2013*; *Ropp et al., 2000*; *Sehgal et al., 2006*; *Suppiah et al., 2011*) and if present, whether this region has viral protein or host cellular protein targets. Evidence supporting and refuting these activities have been building for almost three decades; however, the possibility remains that this region may exert orthogonal activities.

Full characterization of ORF1's functions as well as HEV's full replication cycle have been hampered by a dearth of structural information of the ORF1 protein. Though two structures of small regions within ORF1 have been recently obtained - one being the amphipathic 'thumb' of the RdRp (*Oechslin et al., 2022*), the other being an intra-ORF1 region spanning portions of the pPCP and hypervariable region (HVR; *Proudfoot et al., 2019*) - neither are functional outside of the context of the ORF1 protein, limiting our understanding of how uncharacterized regions of ORF1 fold and operate. Without structural information readily available, *in silico* analyses have been previously attempted to glean information of ORF1's functions, with limited success. For instance, bioinformatic analysis of the HEV pPCP of genotype 1 SAR55 strain predicted three disulfide bridges and a putative zinc-binding motif (*Parvez and Khan, 2014*; *Saraswat et al., 2019*), though previous computational approaches have been lacking in power and iterative ability. With the advent of AlphaFold (*Jumper et al., 2021*; *Tunyasuvunakool et al., 2021*), research into protein structure has entered a new era where testable hypotheses can be generated in an iterative, high-throughput manner.

To understand how the pPCP - and ORF1 more broadly - operate, we opted to combine biochemical, genetic, mass spectrometric, and computational approaches. We identified amino acid motifs within the pPCP that are vital for viral replication via an unbiased triplet alanine scanning mutagenesis, as well as characterized the necessity or dispensability of eight conserved cysteines within the pPCP domain. Of these eight, six were identified as indispensable, forming a hexa-cysteine motif commonly seen in host metal-binding proteins. We established a transcomplementation system to demonstrate that the pPCP is only functional within the context of the full-length (FL) ORF1 protein. We have been able to validate A.I. driven protein structure prediction programs with testable genetic and biochemical data; using the AlphaFold algorithm (*Jumper et al., 2021*), we determined the replicative capacity lost via site-directed mutagenesis to not be due to a deficiency of proteolytic activity, but rather a loss in structural integrity due to an inability to bind divalent metal ions. Moreover, we identified a novel interdomain divalent metal ion binding interaction between the pPCP and the upstream uncharacterized Y-domain of HEV ORF1. Furthermore, utilizing a tolerable epitope locus within ORF1's HVR (*Szkolnicka et al., 2019*), we were able to purify ORF1 protein for downstream inductively coupled plasma mass spectrometry (ICP-MS) analysis and discovered that point mutants in either the pPCP or Y-domain differed in divalent ion binding activity. Taken together, our work demonstrates that HEV ORF1 likely functions as one large multidomain protein that does not undergo proteolytic processing, and that the putative catalytic residues predicted by prior bioinformatic analyses are actually structural in nature via their ability to bind divalent metal ions.

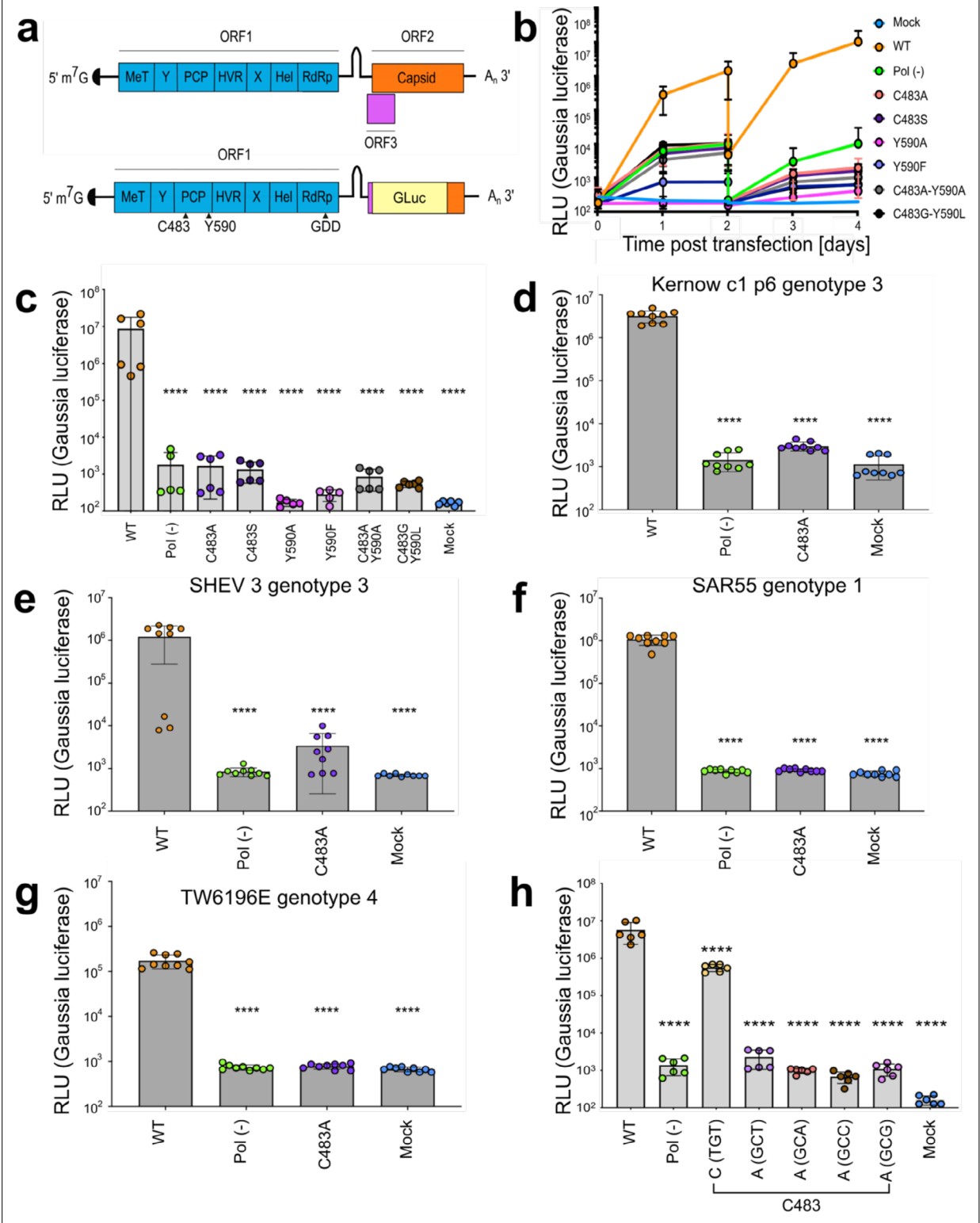

**Figure 1.** Mutations within the hepatitis E virus (HEV) putative protease domain render the virus replication incompetent. (**a**) Genome organization of HEV and Kernow strain genotype 3 reporter replicon. ORF2 and ORF3 were replaced by a *Gaussia* Luciferase reporter in frame with the subgenomic promoter and translation start site. MeT – methyltransferase; Y – Y-domain; PCP – putative papain-like cysteine protease; HVR – hypervariable region; X – macro-domain; Hel – helicase; RdRp – RNA-dependent RNA polymerase. The putative catalytic dyad of HEV within the reporter replicon pPCP is denoted (residues C483 and Y590). GDD – catalytic triad of RdRp. Gluc – Gaussia Luciferase. (**b**), Replication kinetics of Kc1/p6 WT HEV Gluc RNA or HEV pol (-) Gluc, or HEV RNA bearing mutations in the putative PCP transfected into HUH7.5 cells. Cell culture supernatants were collected from

*Figure 1 continued on next page*

*Figure 1 continued*

transfected cells at time points indicated (drop in signal at D2 indicates wash step to eliminate signal from input RNA). (**c**) End point analysis (day 4) of data from panel b. (**d–g**) End point analysis (4 days post transfection) comparison of C483A mutant replication kinetics to WT and pol (-) in (**d**), Kernow strain (genotype 3) (**e**), SHEV (genotype 3) strain (**f**), SAR55 (genotype 1) and (**g**), TW6196E (genotype 4) of HEV when transfected into HepG2C3A human hepatoma cells. (**h**) Alternate cysteine and alanine codons at C483 show viral replication deficiency caused by amino acid substitution and not RNA base substitution. These data represent an end point analysis (4 days post transfection) when HEV RNA replicons are transfected into HUH7 human hepatoma cells. One-way ANOVA with Dunnett's multiple comparison analysis were conducted to determine significance. * - $p<0.05$, ** - $p<0.01$, *** - $p<0.001$, and **** - $p<0.0001$. Data shown in **d–g** are from experiments done in technical and biological triplicate. Data shown in **b**, **c**, and **h** are from experiments done in technical triplicate and biological duplicate. Raw Gluc data provided in file *Figure 1—source data 1*.

The online version of this article includes the following source data and figure supplement(s) for figure 1:

**Source data 1.** Mutations within the hepatitis E virus (HEV) putative protease domain render the virus replication incompetent.

**Figure supplement 1.** Alignment of one or more representative strain(s) from each known hepatitis E virus (HEV) genotype putative papain-like cysteine protease (PCP) domain with rubella virus (RUBV) protease reveals highly conserved octa-cysteine motif in HEV.

## Results

## Mutations within the HEV putative protease domain render the virus replication incompetent

The functional domains and genome organization of HEV (*Figure 1A*) were first suggested based on bioinformatic alignments of the genotype 1 Burma reference strain with other well-characterized viruses in 1992 (*Koonin et al., 1992*), and the putative protease domain was proposed based on limited sequence identity with the distantly related rubella virus. These analyses suggested the existence of a putative papain-like protease within HEV ORF1 and that the proposed catalytic dyad were residues C483 (which is highly conserved across all known HEV genotypes) and H590 - which is variable across all eight known HEV genotypes (*Figure 1—figure supplement 1*). Since, tools have been developed to systematically interrogate the HEV genome, such as the development of infectious clones of cell-culture adapted strains (*Johne et al., 2014*; *Shukla et al., 2012*; *Shukla et al., 2011*), and reporter replicons utilizing green fluorescent protein (GFP; *Emerson et al., 2004*) or Gaussia luciferase (Gluc; *Graff et al., 2005*; *Figure 1A*).

To determine the importance of the residues that have been proposed as the putative catalytic dyad, we mutated C483 and/or Y590 to chemically similar amino acids, alanine, or in the case of Y590, residues found in other genotypes of HEV. Huh7.5 human hepatoma cells were transfected with in vitro transcribed RNA from a recombinant version of an HEV genome derived from the KernowC1/p6 strain (*Shukla et al., 2011*) in which ORF2 and ORF3 are replaced by a secreted version of Gluc (*Shukla et al., 2011*), termed Kc1/p6 Gluc (*Figure 1A*). Gluc activity as a measure for the efficiency of RdRp-mediated viral replication was quantified in the culture supernatants over 4 days post RNA transfection (4 d.p.t.). Transfection of the wild type (wt) Kc1/p6 Gluc into naïve Huh7.5 human hepatoma cells led to a ca. 34,000-fold increase in luminescence over mock cells. Transfection of a polymerase deficient genome harboring a mutation in the highly conserved catalytic triad (GDD motif of the RdRp (deemed pol [-])) expectedly did not augment Gluc activity (*Figure 1B–C*). Notably, genomes harboring mutations in the C483 and/or Y590 positions were incapable of establishing stable replication (*Figure 1B–C*).

We next sought to understand if this lack of viral replication brought on by mutating the highly conserved C483 was unique to the Kc1/p6 cell culture adapted strain of HEV, or if it translated to other known human-tropic HEV strains. Thus, we mutated the C483 residues in the Gluc reporter genome configurations of HEV strains SAR55 (genotype 1), SHEV3 (genotype 3), and TW6196E (genotype 4; *Ding et al., 2018b*), and transfected in vitro transcribed RNA into HepG2C3A human hepatoma cells similar to *Figure 1B–C*. In line with our observations using Kc1/p6 replicon, HEV RNA replication was severely impaired in SAR55, SHEV3, and TW6196E genomes harboring the C483A mutation (*Figure 1D–G*). Notably, Gluc levels were equivalently as low as the pol (-) versions of the reporter replicons.

To determine if this deficiency was due to a disruption in RNA folding, we mutagenized C483 into each available codon for cysteine and alanine (*Figure 1H*). These analyses demonstrated that encoding the alternate cysteine had a mild negative affect on viral replication efficiency, while any alanine codon usage brought replication levels down to those of the Pol (-) mutant (*Figure 1H*), suggesting that the

deficiency primarily lies in protein folding or function. Collectively, these data demonstrate the necessity of these residues for viral fitness, despite Y590 being heterogeneous across HEV viral genotypes.

## HEV ORF1 pPCP cannot function outside of the context of the FL protein, and C483A replication deficiency is rescuable in trans

To probe further the mechanism underlying the functional impairments of the C483 mutants, we devised an experimental system in which HEV RNA replication is uncoupled from protein translation. Following our previously established successful transcomplementation approach for studying HEV ORF3's viroporin function (*Ding et al., 2017*) and cis-regulatory elements responsible for regulating transcription of the subgenomic RNA (*Ding et al., 2018b*), we lentivirally expressed a wt, pol (-), or C483A version of ORF1, or the pPCP alone in HepG2C3A human hepatoma cells (*Figure 2A*). These cells were subsequently transfected with in vitro transcribed RNA from Kernow C1/p6 Gluc wt, pol (-), or C483A genome (*Figure 2A*). Gluc activity as a measure of the efficiency for HEV replication was quantified in the culture supernatants over 4 d.p.t. Notably, when the mock signal fold change over WT luciferase signal was examined, cells expressing a mutant form of ORF1 demonstrate a deleterious effect on WT replicon replication, likely due to competitive inhibitory binding of the mutant protein with the replicon RNA (*Figure 2B*). Impairments in viral genome replication due to the pol (-) or C483A mutations could be rescued in trans by expression of WT ORF1 near to levels of those following transfection of WT replicon RNA (*Figure 2D–E*). Of note, expression of the pPCP failed to restore replication of Kc1/p6 Gluc C483A, suggesting that the functions of this region of ORF1 are not adequately maintained outside of the context of the ORF1 polyprotein. This transcomplementation platform provides further means to uncouple the putative functions of the pPCP, or larger intramolecular regions of ORF1, e.g., polyprotein processing or modulation of the host cellular environment, from viral genome replication.

## Point mutations of highly conserved cysteines and alanine scanning mutagenesis within the pPCP identify residues and regions indispensable for viral replication

We examined the pPCP sequences across all eight known HEV genotypes (*Figure 1—figure supplement 1*) and noticed an octa-cysteine motif highly conserved across all HEV genotypes (*Figure 3A*). Previous work has shown the core hexa-cysteine motif encompassing cysteines 457–483 in SAR55 genotype 1 HEV were necessary for viral replication (*Parvez, 2013*). To determine the necessity of each cysteine to the viral replication cycle in the Kc1/p6 strain of HEV, we mutated each in turn to alanine within the Kc1/p6 Gluc reporter replicon and quantified the luciferase signal 4 d.p.t. (*Figure 3B*). We noticed that of the eight conserved cysteines, only the core six that form a $CxC[x_{11}]CC[x_8]CxC$ hexa-cysteine motif are vital for viral replication, with the first cysteine at position C434 being completely dispensable for replication, and the final cysteine at position C563 having a slight detriment to replication when mutated to alanine (*Figure 3B*).

To determine more broadly which other regions of the HEV pPCP are indispensable for viral replication, we sought to conduct an unbiased genetic mutagenesis screen of the entire pPCP region of 160 amino acids. Site-directed triplicate alanine scanning mutagenesis was conducted across the entirety of the HEV pPCP, identifying several triplicates that offer pro-viral activity, as well as identifying the majority of triplicates vital for viral replicative fitness (*Figure 3C*). Notably, triplicates containing any of the conserved cysteines, as well as the variable amino acid at position 590 were indispensable for viral replication, despite point mutations at positions C434 and C563 being tolerated (*Figure 3B*). Additionally, the amino acid stretches 55–63 and 73–81 directly downstream of the hexa-cysteine motif tolerate mutagenesis quite well, suggesting a possible structural/linker function of these amino acids; these triplicates are not highly conserved across the eight HEV genotypes (*Figure 1—figure supplement 1*).

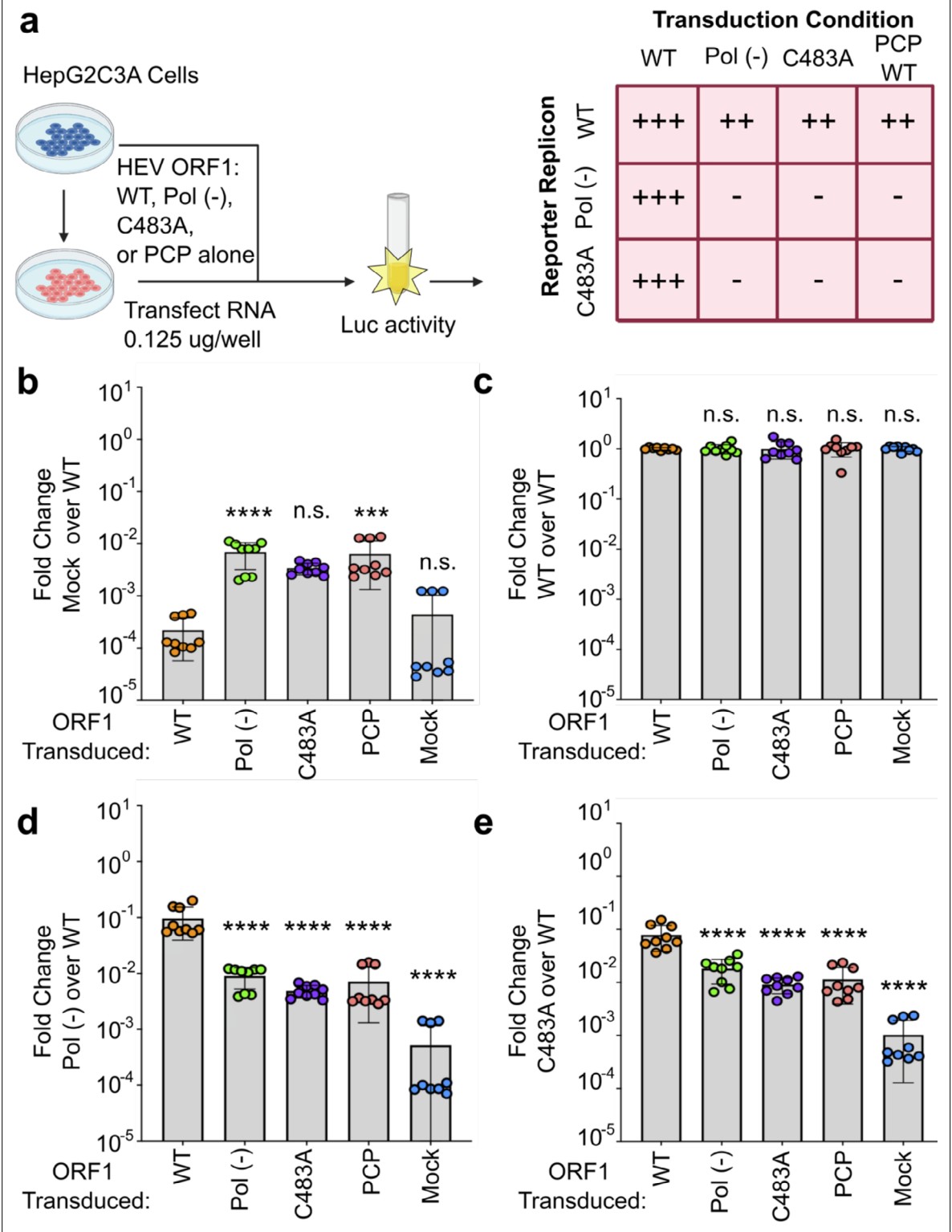

**Figure 2.** Hepatitis E virus (HEV) ORF1 putative papain-like cysteine protease (pPCP) cannot function outside of the context of the full-length protein, and C483A replication deficiency can be rescued in trans. (**a**) Schematic of the transcomplementation assay. HepG2C3A human hepatoma cells were transduced with lentivirus expressing HEV ORF1 wt, Pol (-), C483A, or pPCP only (ORF1 AAs 433–592 in Kc1/p6) and subsequently transfected with in vitro transcribed RNA of wt, pol (-), or C483A replicons. End point analysis was conducted 4 days post transfection with *Gaussia* luciferase quantification. Cells transduced with WT ORF1 can rescue luciferase expression in pol (-) and C483A mutants in trans, whereas all other conditions, including the pPCP out of context of ORF1, cannot. (**b–e**) End point analysis of fold change luciferase signal over WT replicon signal of (**b**), Mock (**c**), WT (**d**), Pol

*Figure 2 continued on next page*

*Figure 2 continued*

(-), or (**e**), C483A. One-way ANOVA with Dunnett's multiple comparison analysis were conducted to determine significance. * - p<0.05, ** - p<0.01, *** - p<0.001, and **** - p<0.0001. Schematic in panel a generated with BioRender. Pol (-) – replication incompetent replicon due to mutation in RNA dependent RNA polymerase. Gluc – *Gaussia* luciferase. Data shown in **b–e** are from experiments done in technical and biological triplicate. Raw Gluc data provided in file *Figure 2—source data 1*.

The online version of this article includes the following source data for figure 2:

**Source data 1.** Hepatitis E virus (HEV) ORF1 putative papain-like cysteine protease (pPCP) cannot function outside of the context of the full-length protein, and C483A replication deficiency can be rescued in trans.

## Hexa-cysteine motif ($CxC[x]_{11}CC[x]_8CxC$) within HEV Kernow pPCP vital for viral replication shares homology with host divalent metal ion binding proteins

Interrogating the results of point mutations to the eight highly conserved cysteines within the pPCP led us to further bioinformatic analysis to determine the function of the vital hexa-cysteine motif. Utilizing motif searches with ScanProSite (*de Castro et al., 2006*) and TrEMBLE (*Bairoch and Apweiler, 2000*; *Boeckmann et al., 2003*; *O'Donovan et al., 2002*), we began by searching for a relaxed expression of the HEV hexa-cysteine motif ($CxC[x]_{3–20}CC[x]_{3–20}CxC$, where x can be any amino acid). From this analysis over 33,000 proteins emerged. To further refine our approach, we identified proteins that matched the HEV CxC motif exactly ($CxC[x]_{11}CC[x]_8CxC$, hereafter referred to as the HEV motif); however, all of the protein hits that emerged are as of now uncharacterized, offering little insight as to the function of this motif (*Figure 3—figure supplement 1A*). Relaxing the criteria to (±) 1 for each of the stretches of [x] ($CxC[x]_{(10–12)}CC[x]_{(7–9)}CxC$) brought forth 26 proteins with known functions (*Figure 3—figure supplement 1A*), enriched for proteins with divalent metal ion binding activity (*Figure 3—figure supplement 1B*). We hypothesized that this region within the pPCP is necessary for metal ion coordination.

## Structural prediction models of HEV pPCP demonstrates low-confidence scores, suggesting lack of highly ordered secondary structure

The dearth of structural information of the HEV ORF1 protein has hampered the complete understanding of the viral replication cycle. To glean more information about the domain organization and protein folding of HEV ORF1, we turned to AlphaFold (*Jumper et al., 2021*) to predict the structure of ORF1. The complete sequence of HEV ORF1 was fed into the AlphaFold algorithm, and the best ranked model (*Figure 4A*) was chosen for further analysis. To gain confidence in the best ranked model, we analyzed the AlphaFold prediction in several ways. First, we analyzed the confidence levels produced by AlphaFold (predicted local distance difference test (pLDDT) score) for each residue across the ORF1 structure prediction (*Figure 4B–C*). This analysis revealed varying levels of confidence throughout the entirety of ORF1, and importantly, low confidence throughout much of the pPCP. We further looked at how AlphaFold predicts the pPCP outside of the context of ORF1 and found that the pLDDT averages are very similar (*Figure 4C*), with the pPCP alone averaging a pLDDT score of 65.92, and the pPCP within the context of ORF1 scoring a slightly better average of 66.05. Second, we tested how closely the AlphaFold prediction aligned with two separate solved structures of fragments of HEV ORF1, as well as the known solved structures of the macro domains in other distantly related viruses. To this end, we used a combinatorial approach of sequence-based alignments with structure-based alignments to gain the most accurate distance matrices of relevant atomic coordinates within each alignment of regions of ORF1 and the corresponding known structures (approach reviewed in *Carpentier et al., 2019*). To accomplish this, we took the ORF1 structure prediction, and using the tool TM-Align (*Zhang and Skolnick, 2005*) we aligned: a region of HEV ORF1 that spans parts of the pPCP and HVR (AAs 510–691) of the SAR55 strain of genotype 1 HEV (PDB: 6NU9; *Proudfoot et al., 2019*), the amphipathic 'thumb' of HEV genotype 3 strain 83-2-27 RdRp (amino acids 1628–1647; ORF1 kc1/p6 residues 1684–1709; *Oechslin et al., 2022*), and the macro domains of Sindbis virus (SINV amino acids 1342–1509 of PDB: 4GUA; *Shin et al., 2012*), and Chikungunya virus (CHIKV; PDB: 3GPG; *Malet et al., 2009*; *Figure 4—figure supplement 1A–C*, respectively).

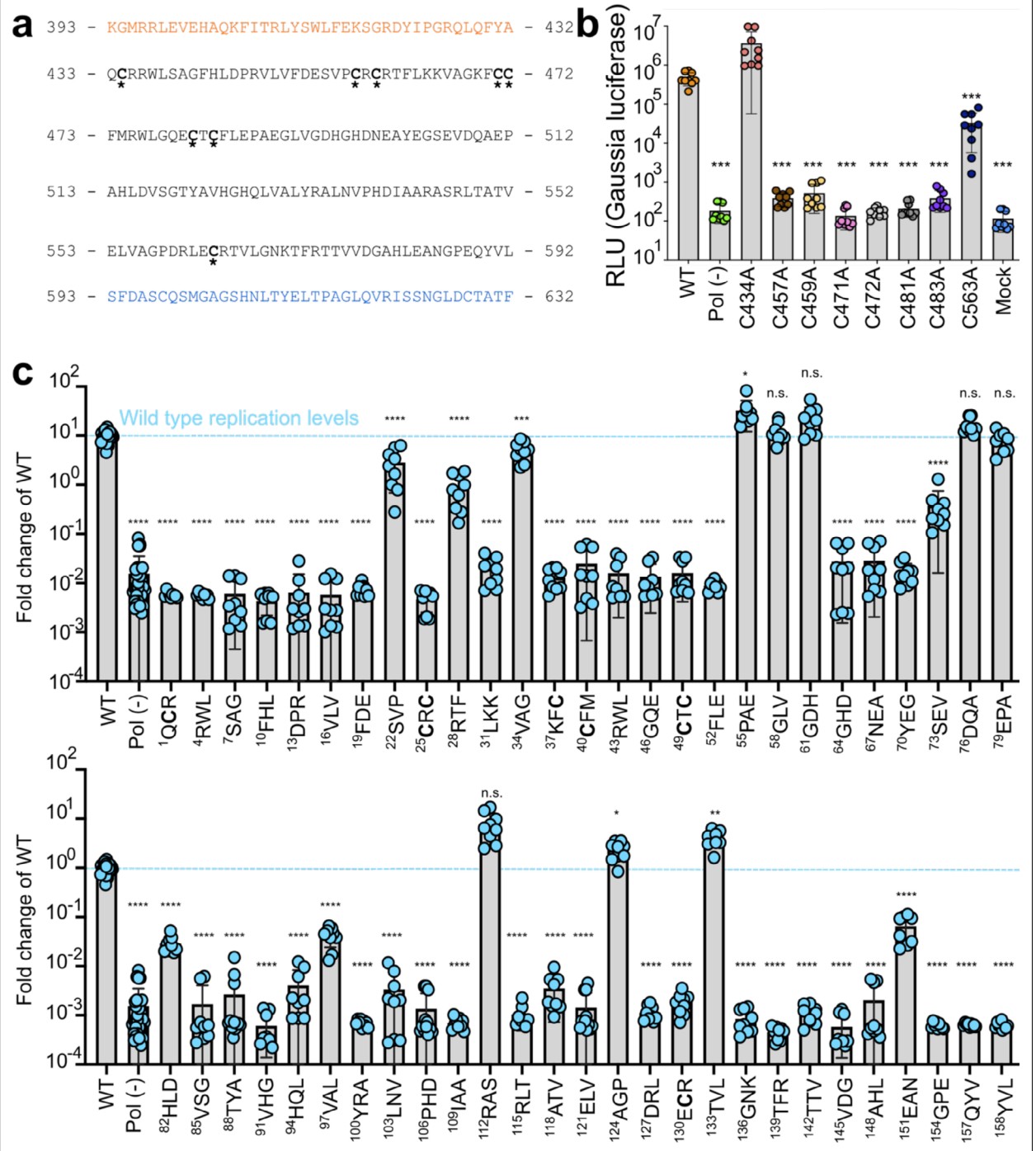

**Figure 3.** Point mutations of highly conserved cysteines and alanine scanning mutagenesis within putative papain-like cysteine protease (pPCP) identifies residues and regions indispensable for viral replication. (**a**) Partial sequence of Kc1/p6 hepatitis E virus (HEV) ORF1 including pPCP that shows eight cysteines that are highly conserved across all eight known HEV genotypes. Orange – upstream 40 amino acids within Y domain prior to pPCP. Blue – downstream 40 amino acids within HVR after pPCP. (**b**), HepG2C3A cells were transfected with in vitro transcribed RNA of WT, polymerase deficient, or point mutants in one of the conserved cysteines within the pPCP. Cell culture supernatants were collected for 4 days post transfection prior to *Gaussia* luciferase (Gluc) quantification. (**c**) Unbiased triplet alanine scanning mutagenesis of entire pPCP region. HepG2C3A cells were transfected with in vitro transcribed RNA of triplet alanine scanning mutant replicons to assess viral replication capacity. Data shown are fold change of wild type replicon. Brown-Forsythe one-way ANOVA with Dunnett's T3 multiple comparison analysis were conducted to determine significance. * - $p<0.05$, ** - $p<0.01$, *** - $p<0.001$, and **** - $p<0.0001$. HVR – hypervariable region. Raw Gluc data provided in file *Figure 3—source data 1*.

The online version of this article includes the following source data and figure supplement(s) for figure 3:

*Figure 3 continued on next page*

*Figure 3 continued*

**Source data 1.** Point mutations of highly conserved cysteines and alanine scanning mutagenesis within putative papain-like cysteine protease (pPCP) identifies residues and regions indispensable for viral replication.

**Figure supplement 1.** Bioinformatic analysis (PROSITE) predicts CxC[X11]CC[X8]CxC motif (CxC motif) to be necessary for divalent metal ion coordination.

Alignment of 6NU9 with the corresponding region of Kc1/p6 ORF1 shows high local alignment identity, with the average distance of this superimposition aligning at 0.67 Å (*Figure 4—figure supplement 1A*). Alignment of the amphipathic RdRp 'thumb' domain similarly shows incredibly high local alignment identity with a 0.48 Å average differential (*Figure 4—figure supplement 1B*). Understandably, alignment replication of the macrodomains of SINV and CHIKV also show high local alignment, though less robust than that of other HEV strains to Kc1/p6 ORF1 of HEV. Notably, the HEV macro domain and the macro domains of SINV and CHIKV share little amino acid similarity and identity, with the Kc1/p6 HEV

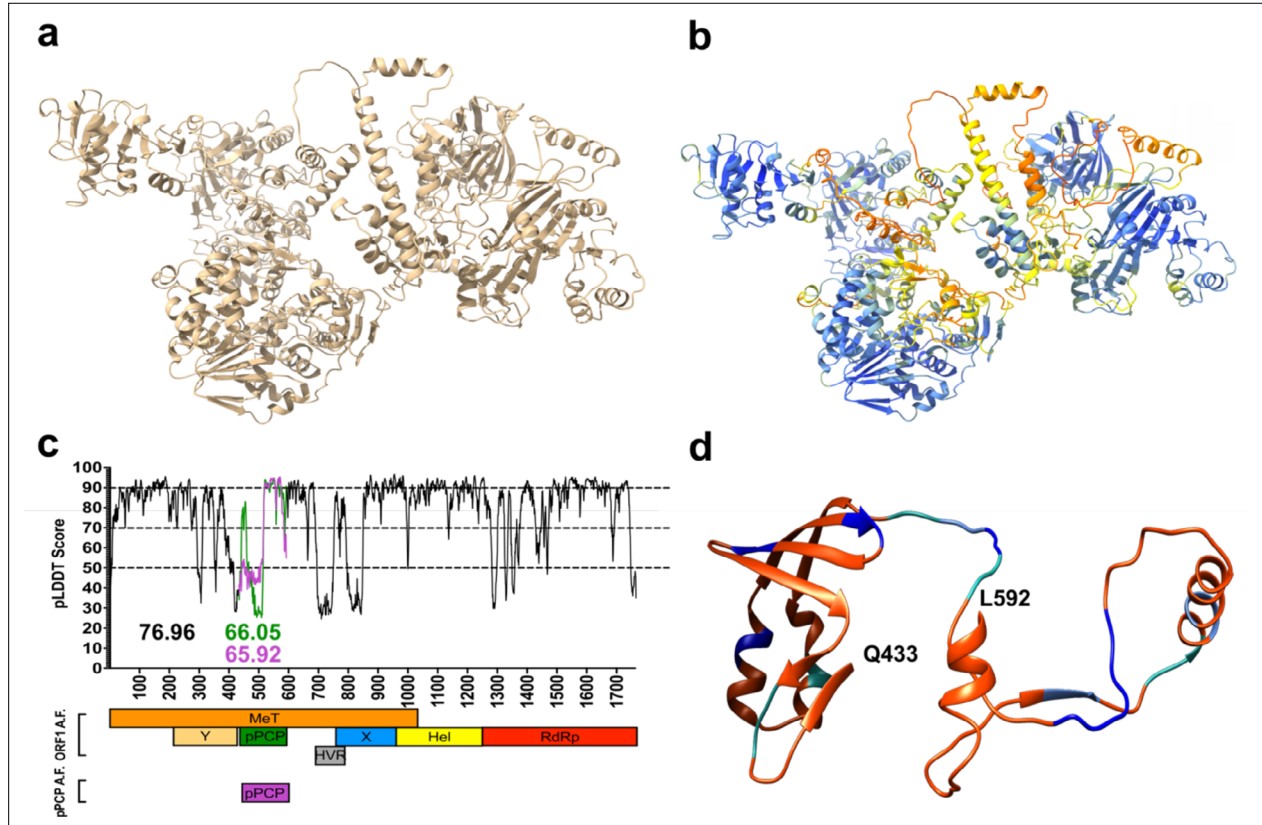

**Figure 4.** Structural prediction models of hepatitis E virus (HEV) putative papain-like cysteine protease (pPCP) demonstrates low-confidence scores, suggesting lack of highly ordered secondary structure. (**a**) HEV ORF1-WT AlphaFold structure prediction. (**b**) HEV ORF1-WT AlphaFold structure prediction pseudo-colored by pLDDT score gradation (darker blue – higher pLDDT Score, darker red – lower pLDDT score). (**c**) pLDDT score of AlphaFold prediction of HEV ORF1 across HEV genome organization for all of ORF1 (black, average 76.96), pPCP when measured with entirety of ORF1 (green, average 66.05), and pPCP when predicted by AlphaFold alone (purple, average 65.92). (**d**) HEV ORF1 pPCP AlphaFold prediction pseudo-colored by alanine scanning mutagenesis ORF1 replication tolerance. Color based on fold change of WT in (*Figure 3C*). Orange – below (–2) Sea green – between (–2) and (–1). Cornflower blue – between (–1) and 0. Dark blue – replicated above WT levels. Beginning and end residues of pPCP noted in bold.

The online version of this article includes the following source data and figure supplement(s) for figure 4:

**Figure supplement 1.** AlphaFold predicts structured domains of hepatitis E virus (HEV) ORF1 and viral protease of hepatitis A virus (HAV) with high confidence.

**Figure supplement 2.** Sequence alignment of Kc1/p6 macrodomain with macrodomains of Sindbis virus (SINV) and Chikungunya virus (CHIKV).

**Figure supplement 2—source data 1.** Similarity and identity percentages of Kc1/p6 macrodomain with macrodomains of Sindbis virus (SINV) and Chikungunya virus (CHIKV).

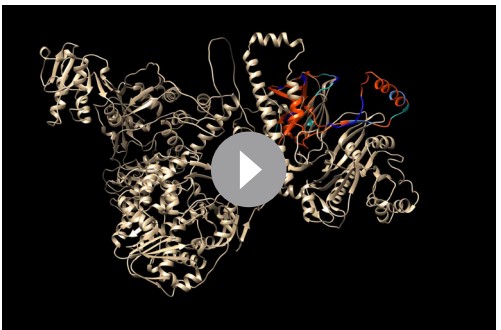

**Video 1.** AlphaFold Prediction of ORF1 WT - putative papain-like cysteine protease (pPCP) Alanine Scan Gradation. Best ranked model of ORF1 WT AlphaFold prediction algorithm that transitions to a visual representation of the alanine scanning mutagenesis data across the pPCP. Color gradation corresponds fold change of WT of alanine triplet. Below (–2) fold change is orange. Between (–2) and (–1) fold change is sea green. Between (–1) and 0 fold change is cornflower blue, Above WT replication levels is dark blue.
https://elifesciences.org/articles/80529/figures#video1

macro domain sharing only 36.45% sequence identity and 54.21% sequence similarity with the SINV macro domain, and sharing only 35.58% sequence identity and 56.73% sequence similarity with the CHIKV macrodomain (*Figure 4—figure supplement 2*; *Figure 4—figure supplement 2— source data 1*). Despite this sequence disparity, these viral macrodomains share high structural identity, with the ORF1 and SINV superimposition aligning at 1.79 Å on average, and similarly, aligning with the CHIKV superimposition at 1.89 Å on average (*Figure 4—figure supplement 1C*).

Importantly, these aforementioned highly ordered secondary structures within HEV ORF1 all exhibit much higher AlphaFold pLDDT scores on average than the whole of the pPCP (*Figure 4C*). This prompts several considerations: (1) Alpha-Fold is not able to accurately predict the amino acid backbone or side chains of the pPCP with a high amount of confidence. (2) If this region were a protease domain, it would likely have a highly ordered and readily predicted secondary structure. To test this latter point, we analyzed Alpha-Fold's ability to accurately predict the structure of a known protease, which it does very well. When comparing the known structure of the hepatitis A virus (HAV) 3 C protease (*Bergmann et al., 1999*) with the AlphaFold prediction, they align with an average superimposition distance of 1.03 Å (*Figure 4—figure supplement 1D*). Interestingly, Alpha-Fold was able to fold the HAV protease with very high confidence, with an average pLDDT score of 97.14 across the entirety of the prediction, giving high confidence of interpretation to both the amino acid backbones and side chain chains (*Figure 4—figure supplement 1D*). Due to the low pLDDT score of the pPCP, we visualized the predicted folding of this region by color-coding the outcomes of the alanine-scanning mutagenesis to better understand whether the secondary structures predicted by AlphaFold stand. We found that the majority of the predicted alpha-helices and beta-sheets do not tolerate mutations well, while the majority of the tolerated mutants lie within predicted regions of disorder (*Figure 4D*; *Video 1*). Taken together, these results suggest the structure predictions of HEV ORF1 by AlphaFold demonstrate high confidence within known regions that possess a high level of secondary structure. AlphaFold is able to predict the structure of a known viral protease with high confidence, suggesting further that the HEV pPCP domain does not possess the necessary secondary structure of a protease. This prediction of the HEV ORF1 pPCP is bolstered by the enrichment of replication tolerant triplets from the alanine scanning mutagenesis to preferentially localize to predicted areas of disorder (*Figure 4D*; *Video 1*), suggesting as a whole that the pPCP does not fold as a protease with a catalytic pocket.

## Structural prediction models suggest that mutating cysteines within pPCP disrupts divalent ion coordination pockets and novel domain-domain interaction with upstream Y-domain

Upon demonstrating that mutations within the pPCP domain prevent HEV from replicating, we sought to further elucidate potential mechanisms by which this deficiency occurs. Utilizing the structural predictions of HEV ORF1 generated with AlphaFold, we began by analyzing the predicted folding structure of WT ORF1, and two of the point mutants within the pPCP: C483A and C563A (chosen based on the heterogeneity of their phenotypes). C483A is fully replication deficient, while C563A is blunted in replication at two orders of magnitude lower than WT 4 d.p.t. (*Figure 3B*).

We next fed the sequences of the C483A and C563A mutant sequences of ORF1 into the AlphaFold algorithm. Upon inspection of the best ranked models for each of these versions of ORF1 (WT, C483A, and C563A), we noticed a novel pseudo-zinc-finger formed by the amino acids underlying the HEV

hexa-cysteine motif, despite the low pLDDT scores within this region (*Figure 5A*, left; *Videos 2–4*, respectively). Mutation of C483A, but not C563A, is predicted to disrupt the molecular architecture of this pseudo-zinc-finger; however, mutation of either causes a relaxation of several predicted bond lengths to beyond biological relevance (*Zheng et al., 2008*; *Figure 5—figure supplement 1*, *Videos 2–3*). Cysteine side chain residues at positions 457 and 459 at the end of the predicted beta sheet leading into the pseudo-zinc-finger were shown to be projecting into inter-domain space. These residues are predicted to form a potential tetrahedral divalent ion binding pocket (*Laitaoja et al., 2013*) with D248 and H249 in the upstream Y-domain (*Figure 5A*, middle), which had pLDDT scores of 82.99 and 87.67, respectively, giving high confidence in the predicted location of the amino acid backbones and potentially their side-chains (*Figure 4C*). Cysteine, histidine, and aspartic acid residues are well known to bind divalent metal ions and form tetrahedral geometry (*Laitaoja et al., 2013*; *Zheng et al., 2008*). To further test whether this predicted interaction between C457, C459, D248, and H249 was vital to viral replication, we mutated either D248 or H249 to alanine in the Kc1/p6 Gluc replicon and quantified Gluc expression 4 d.p.t. D248 was dispensable for viral replication while H249 is not *Figure 5—figure supplement 1*; D248 is variable across all eight known HEV genotypes while H249 is highly conserved (*Figure 5B*). This observation led us to inquire as to what the disruptions these point mutations could have on ORF1 globally. We noticed that by comparing the structure of ORF1 WT (*Video 2*) next to the mutants while highlighting domains of ORF1 with well-defined functions such as the methyltransferase, helicase, and RdRp, structural differences emerge. Mutating C483A, C563A, or H249A are predicted to cause regions of the methyltransferase, helicase, and RdRp to reconfigure (*Figure 5A*, right; *Videos 3 and 4*, 6 respectively). Furthermore, mutating C563A, D248A, or H249A causes a predicted membrane association domain that is exposed in the WT ORF1 protein to become buried, possibly preventing the association with intracellular membranes and preventing/hindering the formation of a replication compartment (*Metzger et al., 2022*; *Szkolnicka et al., 2019*; *Videos 4–6*, respectively). Taken together, these results suggest that by interfering with divalent metal ion binding domains within the pPCP, structural domains vital to viral replication form aberrantly, and prevent HEV ORF1 from efficiently replicating.

## HEV hexa-cysteine motif coordinates biologically relevant divalent metal ions

Outside of our own analysis that suggests the pPCP of HEV ORF1 has metal ion binding activity, other groups have identified regions within ORF1 predicted to harbor $Ca^{2+}$ and $Zn^{2+}$ ion binding sites (*Parvez and Khan, 2014*; *Proudfoot et al., 2019*). However, the low abundance of ORF1 protein within infected cells and the lack of well-characterized ORF1 specific antibodies has hampered attempts at purifying a replication competent ORF1 protein. This barrier has only recently been overcome when tolerant epitope tag insertion sites were discovered within the HVR region of ORF1 and characterized (*Metzger et al., 2022*; *Szkolnicka et al., 2019*).

Utilizing an HA-tag insertion site flanked on either side by the linker sequence (AAAPG-HA-AAPG, hereafter referred to as HA-tagged) within the HVR of ORF1 (*Szkolnicka et al., 2019*), we generated an overexpression system of WT or mutant HA-tagged ORF1 via lentiviral transduction of Huh7 human hepatoma cells (*Figure 6A*). To determine if the tolerable or intolerable mutations within the pPCP at residues D248A or E489A, or at C483A, C563A, and H249A, respectively (*Figure 3C*), or the Pol (-) mutation in the RdRp affected ORF1's ability to bind divalent ions, we immune purified each protein via the HA-tag mediated immunopurification (*Figure 6B*) and subjected each purified protein eluant to ICP-MS (*Figure 6C*; *Figure 6—figure supplement 1*). Element signatures indicate that while each mutant is heterogenous with each other in binding activity for most divalent ions, all mutations within the pPCP bind less $Zn^{2+}$ than WT. Unsurprisingly, the element signature for the Pol (-) mutant bound all elements similar to WT (*Figure 6—figure supplement 1*). These results suggest that divalent ion binding capacity likely affects proper structural confirmation of ORF1, leading to differential replication capacity based on mutation position.

## Loss of divalent ion binding activity leads to differences in subcellular localization of HEV ORF1

To determine whether the differential in divalent ion binding potential due to mutations within the pPCP or upstream Y-domain is responsible for a loss of subcellular localization, we turned to confocal

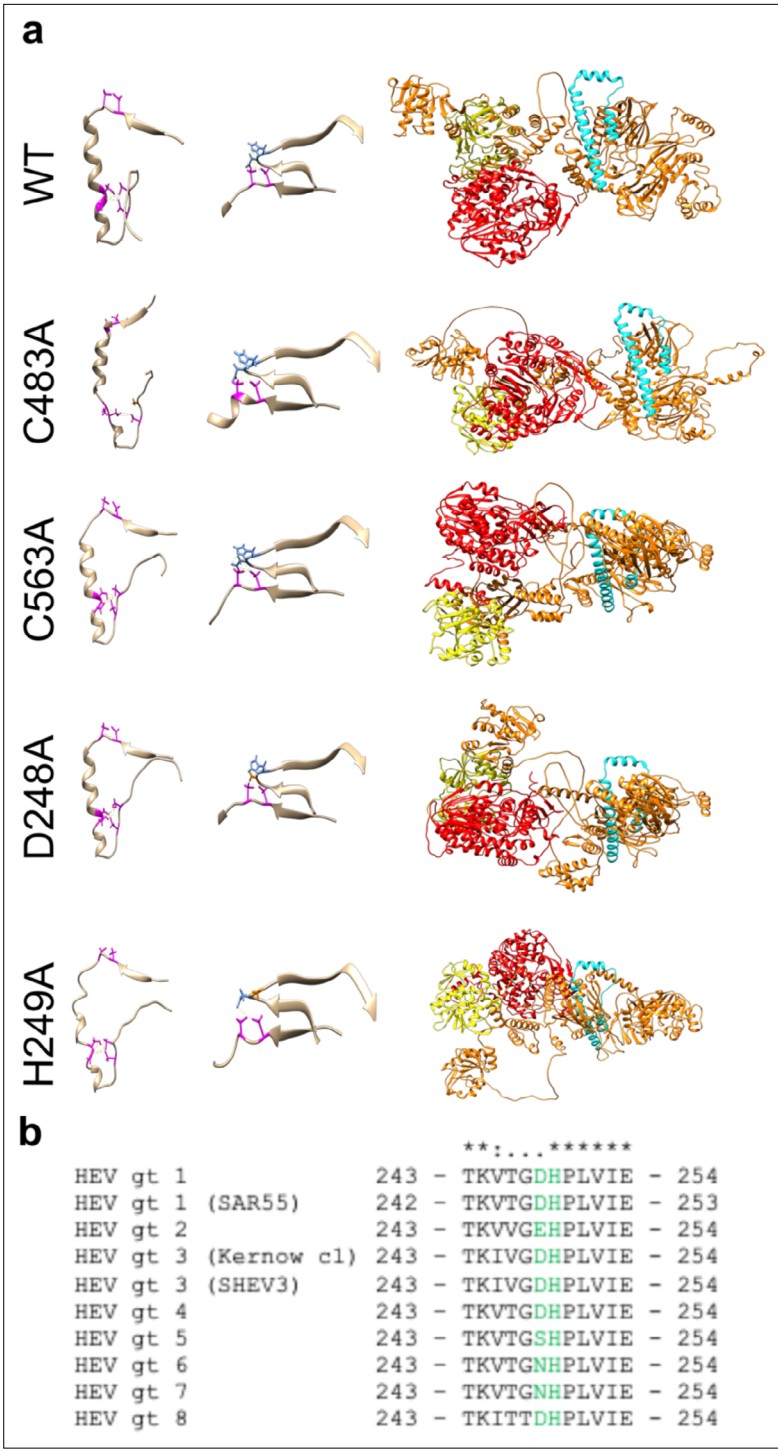

**Figure 5.** Structural prediction models suggest conserved cysteines within CxC[x₁₁]CC[x₈]CxC motif form divalent ion coordination pockets and novel domain-domain interaction with upstream Y-domain. (**a**) Alphafold structural predictions of domains within hepatitis E virus (HEV) ORF1. Left: pseudo zinc-finger (amino acids 451–493 within putative papain-like cysteine protease [pPCP]). Magenta: conserved cysteines C457, C459, C471, C472, C481, and C483A. Potential divalent ion coordination tetrahedron outlined in yellow hatched line. Middle: amino acids 242–259 of HEV ORF1 Y-domain, and amino acids 451–462 of HEV pPCP. Conserved cysteines C457 and C459 are outlined in magenta. D248 and H249 of upstream Y-domain highlighted in blue. Novel interdomain divalent ion coordination domain outlined in yellow hatched line. Right: HEV ORF1 protein demonstrating folding of WT and point mutant proteins. Orange (AAs 1–1036): methyltransferase (*Magden et al., 2001*). Yellow (AAs 1018–1262):

*Figure 5 continued on next page*

*Figure 5 continued*

Helicase (**Devhare et al., 2014**; **Karpe and Lole, 2010**). Red (AAs 1257–1709): RNA dependent RNA polymerase (**Koonin et al., 1992**; **Oechslin et al., 2022**). Cyan: Putative membrane association domain (**Parvez, 2017**). (**b**) Multiple sequence alignment of HEV genotypes 1–8 of partial Y-domain containing variable residue D248 and highly conserved residue H249. (*) identical residue. (:) similar residue. (.) dissimilar residue. Yellow hatched line – bonds between coordinating amino acids.

The online version of this article includes the following figure supplement(s) for figure 5:

**Figure supplement 1.** Replication end point analysis of WT-HA, D248A, and H249A *Gaussia* luciferase (Gluc) constructs.

---

microscopy. Cells bicistronically expressing zsGreen as a marker of transduction as well as WT ORF1 without an epitope tag, WT ORF1-HA-tagged, C483A-HA-tagged, C563A-HA-tagged, D248A-HA-tagged, or H249A-HA-tagged were imaged for zsGreen, and using antibodies against the HA-tag and nuclei were imaged to visualize ORF1 subcellular localization patterning. Cells expressing ORF1 WT-HA showed significant expression of ORF1 in both the cytoplasm and nucleus as previously reported (**Metzger et al., 2022**) and showed many puncta aggregates throughout the cytoplasm (**Figure 7**). In contrast, the replication deficient ORF1 C483A-HA expressing cells lost the ability for ORF1 to localize to the nucleus and was found dispersed throughout the cytoplasm (**Figure 7**). Cells expressing the C563A-HA mutant of ORF1 showed similar localization patterns to ORF1 WT-HA, with puncta forming in the cytoplasm and maintaining the ability to localize to the nucleus (**Figure 7**).

Cells expressing mutations in the novel upstream interacting Y-domain were also varied in their localization when compared to ORF1 WT-HA. Replication competent ORF1 D248A-HA, like C563A-HA, shared localization patterns with ORF1 WT-HA, forming cytoplasmic puncta and localizing within the nucleus (**Figure 7**). In contrast, replication deficient ORF1 H249A-HA showed very disperse cytoplasmic localization, no aggregate forma-tion, and a lesser ability to localize to the nucleus (**Figure 7**). Taken together with the replication data of these epitope-tagged ORF1 mutants

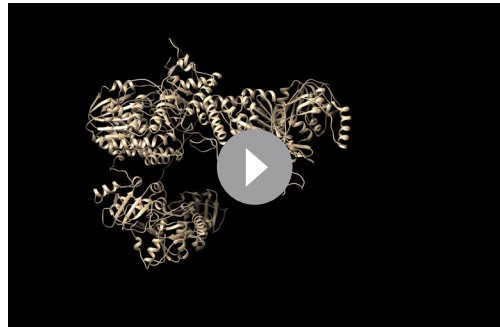

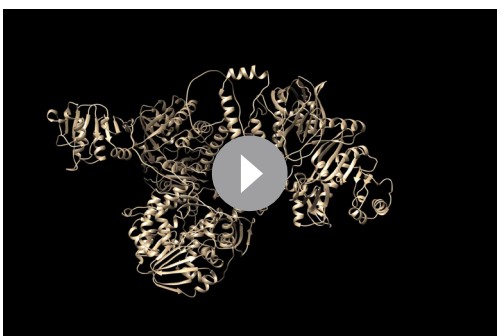

**Video 2.** AlphaFold Prediction of ORF1 WT. Best ranked model of AlphaFold prediction algorithm of ORF1 WT, exhibiting the conserved hexa-cysteine motif (magenta), interacting Y-domain residues D248 and H249 (cornflower blue), methyltransferase domain (orange), putative papain-like cysteine protease (pPCP; green), helicase (yellow), RNA dependent RNA polymerase (red), and putative intracellular membrane association site (cyan). Hatched yellow lines – measured distances between sulfur atoms of cysteines, second oxygen of aspartic acid, or second nitrogen of histidine. Orientation of each video is with the putative membrane association site pointing up, with the pseudo zinc-finger composed of cysteines within the hexa-cysteine motif on the right.
https://elifesciences.org/articles/80529/figures#video2

**Video 3.** AlphaFold Prediction of ORF1 C483A. Best ranked model of AlphaFold prediction algorithm of ORF1 C483A, exhibiting the conserved hexa-cysteine motif (magenta), interacting Y-domain residues D248 and H249 (cornflower blue), methyltransferase domain (orange), putative papain-like cysteine protease (pPCP; green), helicase (yellow), RNA dependent RNA polymerase (red), and putative intracellular membrane association site (cyan). Hatched yellow lines – measured distances between sulfur atoms of cysteines, second oxygen of aspartic acid, or second nitrogen of histidine. Orientation of each video is with the putative membrane association site pointing up, with the pseudo zinc-finger composed of cysteines within the hexa-cysteine motif on the right. Mutation of C483A is predicted and shown here to disrupt the molecular architecture of a pseudo-zinc-finger, likely causing a relaxation of several predicted bond lengths to beyond biological relevance (**Zheng et al., 2008**).
https://elifesciences.org/articles/80529/figures#video3

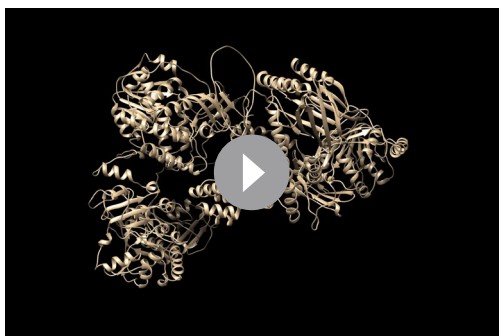

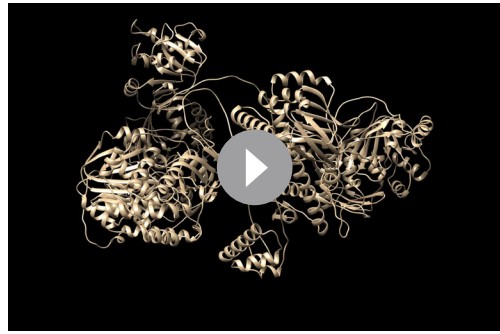

**Video 4.** AlphaFold Prediction of ORF1 C563A. Best ranked model of AlphaFold prediction algorithm of ORF1 C563A, exhibiting the conserved hexa-cysteine motif (magenta), interacting Y-domain residues D248 and H249 (cornflower blue), methyltransferase domain (orange), putative papain-like cysteine protease (pPCP; green), helicase (yellow), RNA dependent RNA polymerase (red), and putative intracellular membrane association site (cyan). Hatched yellow lines – measured distances between sulfur atoms of cysteines, second oxygen of aspartic acid, or second nitrogen of histidine. Orientation of each video is with the putative membrane association site pointing up, with the pseudo zinc-finger composed of cysteines within the hexa-cysteine motif on the right. Mutation of C563A is predicted to cause a relaxation of several predicted bond lengths to beyond biological relevance (*Zheng et al., 2008*).

https://elifesciences.org/articles/80529/figures#video4

**Video 5.** AlphaFold Prediction of ORF1 D248A. Best ranked model of AlphaFold prediction algorithm of ORF1 D248A, exhibiting the conserved hexa-cysteine motif (magenta), interacting Y-domain residues D248 and H249 (cornflower blue), methyltransferase domain (orange), putative papain-like cysteine protease (pPCP; green), helicase (yellow), RNA dependent RNA polymerase (red), and putative intracellular membrane association site (cyan). Hatched yellow lines – measured distances between sulfur atoms of cysteines, second oxygen of aspartic acid, or second nitrogen of histidine. Orientation of each video is with the putative membrane association site pointing up, with the pseudo zinc-finger composed of cysteines within the hexa-cysteine motif on the right.

https://elifesciences.org/articles/80529/figures#video5

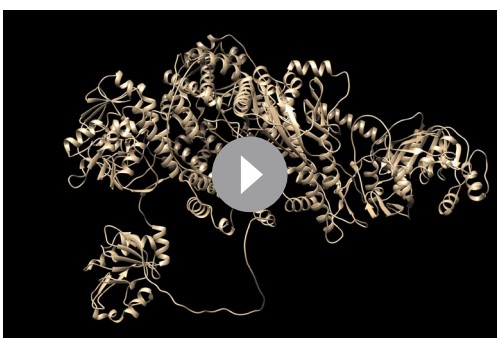

**Video 6.** AlphaFold Prediction of ORF1 H249A. Best ranked model of AlphaFold prediction algorithm of ORF1 H249A, exhibiting the conserved hexa-cysteine motif (magenta), interacting Y-domain residues D248 and H249 (cornflower blue), methyltransferase domain (orange), putative papain-like cysteine protease (pPCP; green), helicase (yellow), RNA dependent RNA polymerase (red), and putative intracellular membrane association site (cyan). Hatched yellow lines – measured distances between sulfur atoms of cysteines, second oxygen of aspartic acid, or second nitrogen of histidine. Orientation of each video is with the putative membrane association site pointing up, with the pseudo zinc-finger composed of cysteines within the hexa-cysteine motif on the right.

https://elifesciences.org/articles/80529/figures#video6

(*Figure 5—figure supplement 1*), localization of ORF1 seems to correlate with replicative capacity, with nuclear localization being lost or diminished in mutants that cannot replicate (C483A-HA and H249A-HA), and mutants able to establish replication (C563A-HA and D248A-HA) share localization patterns with ORF1 WT-HA. These results shed light on a potential novel mechanism by which mutants in the pPCP and upstream Y-domain interfere with HEV's replication cycle.

## Discussion

With the advent of reporter replicons for the HEV replicase in the early 2000s (*Emerson et al., 2004*), research into the functional domains of ORF1 become possible. Perturbations to the viral replicase could be assessed qualitatively and be quantified for the first time, allowing researchers to begin dissecting regions of ORF1 necessary to viral replication. In our study, we first sought to determine whether our results with the Kc1/p6 genotype 3 HEV were in agreement with previous results that utilized a GFP reporter replicon of HEV genotype 1 SAR55 strain in S10-3 cells (*Parvez, 2013*). We were able to demonstrate that mutations in the core six of the eight highly conserved cysteines within the pPCP, as well as

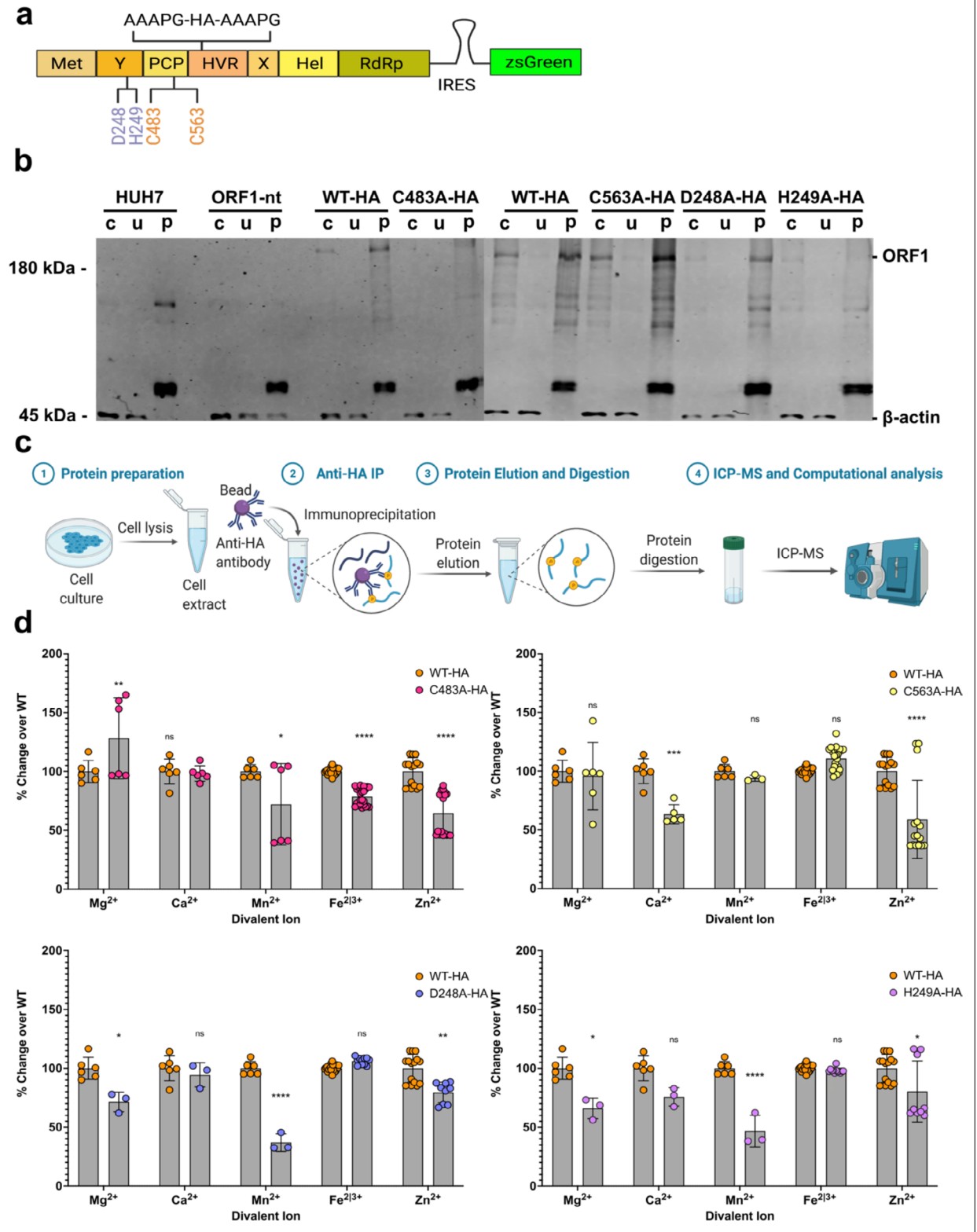

**Figure 6.** Inductively coupled plasma mass spectrometry (ICP-MS) shows divalent cation coordination by hepatitis E virus (HEV) ORF1. (**a**) HA-epitope tag flanked by linker sequence inserted into hypervariable region (HVR) of HEV ORF1 lentiviral construct. (**b**) Western blot analysis of HUH7 human hepatoma cell produced epitope tagged ORF1 purification (crude cell lysate, unbound fraction, purified ORF1). (**c**) Workflow of protein sample preparation, purification, and HNO3/H2O2 digestion for ICP-MS. IP- Immuno-purification WT HA-tagged ORF1 (left), C483A HA-tagged ORF1 (middle), and C563A HA-tagged ORF1 (right). (**d**) Fold-change over WT of biologically relevant divalent metal ions (*Zheng et al., 2008*) bound by purified mutant

*Figure 6 continued on next page*

*Figure 6 continued*

epitope tagged ORF1. Any trace metal not measured as a value above a MilliQ water only purification control was excluded from this analysis on a run-by-run basis due to not being above IP buffer elution conditions. Two-way ANOVA with Dunnett's multiple comparison analysis were conducted to determine significance. * - p<0.05, ** - p<0.01, *** - p<0.001, and **** - p<0.0001. Schematics in **a** and **c** were created with Biorender.com. Unedited western blots included in file *Figure 6—source data 1*.

The online version of this article includes the following source data and figure supplement(s) for figure 6:

**Source data 1.** Inductively coupled plasma mass spectrometry (ICP-MS) shows bivalent cation coordination by hepatitis E virus (HEV) ORF1.

**Figure supplement 1.** Inductively coupled plasma mass spectrometry (ICP-MS) shows divalent cation coordination by hepatitis E virus (HEV) ORF1 disrupted when mutated at residue C483.

the heterogenous residue at position 590, renders Kc1/p6 HEV replication incompetent. We were also able to show that the putative catalytic cysteine in the HEV pPCP renders HEV in three additional genotypes replication incompetent. Taking this analysis further, we demonstrate that the dysfunction in our reporter replicon is likely at the protein level due to the replicon being able to tolerate an alternate codon for cysteine and none of the codons for alanine, suggesting a conserved need of this amino acid residue for HEV.

Of all the domains within ORF1, the functions of the pPCP remain the most debated. Though evidence for and against proteolytic cleavage continues to mount on both sides, it is important to take the scientific results, as well as the functionality of the ORF1 protein, in context. Our data in this study has shown that the pPCP of ORF1 cannot function outside of the context of the FL protein, which is rather uncommon for many RNA viruses such as HAV (*Lemon et al., 1991*), HCV (*Yang et al., 2000*), and flaviviruses such as Zika virus (*Ding et al., 2018a*). While most characterized (+) ssRNA viruses rely on proteases to liberate individual gene products from their encoded polyprotein, HEV may be an exception. While it remains conceivable that host proteases may post-translationally process ORF1, there is rather limited evidence that subunits of ORF1 itself harbors proteolytic activity. Furthermore, if processing were to occur, it is likely that only a small fraction of ORF1 might be cleaved, as suggested previously (*Metzger et al., 2022*); however, the smaller species of ORF1 in the previously cited study were unable to be characterized by mass spectrometric analysis, leaving the processing of ORF1 still subject to debate. The inability for the putative HEV PCP to act outside of the context of the FL ORF1 protein suggests that it has some orthogonal activity and that HEV ORF1 likely functions as one large multi-domain protein. However, other known viral proteases such as HCV's NS3/4A possess *cis*-acting activity as well as *trans*-acting activity (*Kazakov et al., 2015*). With this in mind, the possibility remains that some undiscovered domain of ORF1 that possesses *cis*-acting processing activity is required to functionally rescue a defective genome in trans; results in favor if this possibility have yet to come to light.

To take an unbiased approach to analyzing the pPCP domain, we attempted to identify motifs within the region that were either vital or dispensable to viral replication. To this end, we conducted an alanine scanning mutagenesis screen in triplets across the entire viral region and found that while the majority of the region is needed for replication, there were 11/54 triplets that were able to replicate at near WT levels. Several of these triplets fell very near to the HEV ($CxC[x_{11}]CC[X_8]CxC$) motif within the pPCP; we aimed to identify the potential function of this region, as well as obtain as much structural information. To this end, we were able to identify proteins with known functions that shared close homology to the HEV motif and found that these proteins were enriched for metal ion binding or for disulfide bond formation, suggesting a structural activity rather than a catalytic one. We then capitalized on the power of AlphaFold to gain structural insights into the nature of this vital region. When looking at the structure predictions of the HEV motif, we noticed a striking pseudo-zinc finger formation, as well as a tetrahedral binding pocket canonically associated with $Zn^{2+}$ binding with two residues in the upstream Y-domain. When mutations to conserved cysteines or the aspartic acid or histidine in the upstream Y-domain were modeled, several changes noticeable: first, for the two mutants that are rendered replication incompetent (C483A and H249A), the alpha-helix of the pseudo zinc-finger is disrupted. Furthermore, several bond lengths between potential coordinating residues are relaxed to beyond biological relevance. Furthermore, a putative membrane contact site predicted in *Parvez, 2017* becomes buried in these mutants, hinting at a potential mechanism behind the loss of replicative ability. Many RNA viruses are known to adopt a similar strategy of metal ion coordination to carry out necessary functions (*Chasapis, 2018*). For instance, HCV utilizes a metalloprotein in its replicase,

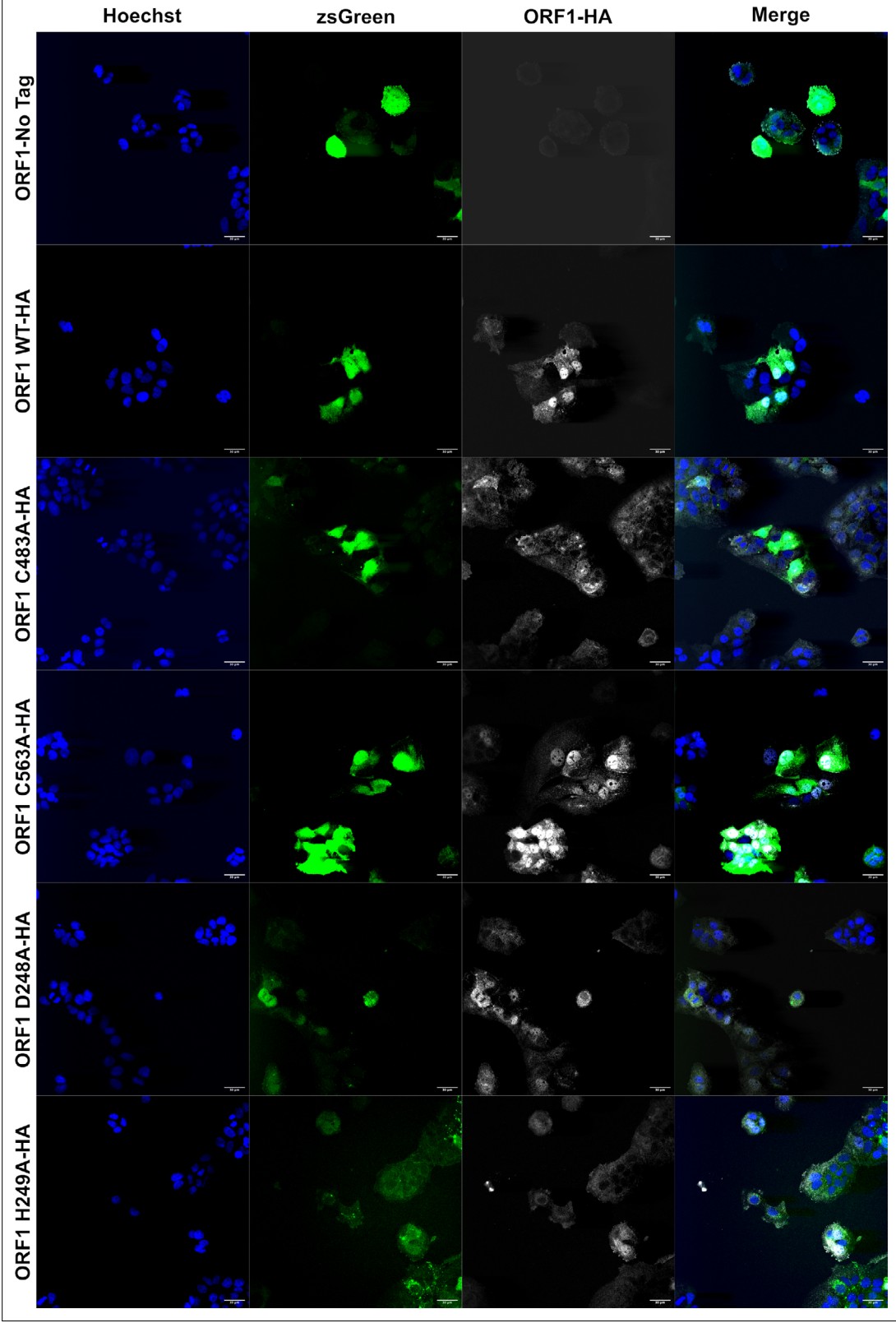

**Figure 7.** Immunofluorescence of epitope tagged ORF1 demonstrates loss of membrane association when divalent ion coordination residues are mutated. Confocal microscopy images of HUH7 cells stably expressing ORF1 bicistronic for zsGreen. ORF1-HA tagged proteins were imaged using a rabbit-anti-HA antibody and AlexaFluor647 (goat anti-rabbit IgG [H+L]) secondary antibody. Nuclei were visualized with Hoechst 33342 stain. All images

*Figure 7 continued on next page*

*Figure 7 continued*

taken at 40× magnification and analyzed using Fiji (ImageJ2) image analysis software. Source files of all TIFF images included in ***Figure 7—source data 1***.

The online version of this article includes the following source data for figure 7:

**Source data 1.** Immunofluorescence of epitope tagged ORF1 demonstrates loss of membrane association when divalent ion coordination residues are mutated.

the nonstructural protein NS5A (***Tellinghuisen et al., 2004***; ***Tellinghuisen et al., 2005***); HCV utilizes four cysteines within a C$[x]_{17}$CxC$[x]_{20}$C motif, conserved among *Hepacivirus* and *Pestivirus* genera, for Zn$^{2+}$ coordination and proper function of the HCV replicase machinery.

To determine whether mutants in these predicted metal ion binding motifs actually led to a decrease in ion binding activity, we needed to be able to purify ORF1 protein for subsequent analyses. One of the many difficulties in interrogating HEV ORF1 is its low expression level in infected cells, as well as the lack of a well-characterized commercial antibody (***Lenggenhager et al., 2017***). However, recent identification of sites within ORF1 that tolerate epitope tags without sacrificing viral replicative capacity have opened up new avenues for researchers to investigate ORF1, and the pPCP, more robustly and critically. Utilizing one such insertion site, we were able to purify WT and mutant ORF1 proteins and subject them to ICP-MS. We found that across all mutants, none were able to bind zinc ion species as well as the WT ORF1, save the Pol(-) RdRp mutant, lending validity to our structural hypotheses generated with AlphaFold. We then determined whether the predicted burying of the putative membrane contact site in the replication deficient mutants affected ORF1 localization within cells expressing the epitope tagged ORF1. Utilizing confocal microscopy, we were able to demonstrate that the C483A mutation loses ORF1 nuclear localization, while the H249A mutation decreases nuclear localization of ORF1, and prevents puncta formation throughout the cytoplasm, which lies in stark contrast to WT ORF1. Taken together, with the advent of new tools such as reporter replicons for HEV, AlphaFold, and tolerable epitope insertion sites discovered within ORF1, our bioinformatic and genetic analyses have been able to go far beyond previous attempts at predicting functional domains within ORF1: we have been able to demonstrate a powerful, iterative pipeline of testing A.I. driven predictions via hypothesis generation and testing with the tools at our disposal. We have been able to demonstrate a novel domain-domain interaction between the upstream Y-domain and the metal-coordinating structural domain of HEV, previously (and incorrectly called the PCP), and we suggest a change in the accepted nomenclature of this vital and enigmatic viral region to reflect this function.

## Materials and methods
### Cell lines and cell culture
HepG2C3A cells (ATCC and CRL-10741), Huh7, and Huh7.5 cells (kindly provided by Charles Rice, The Rockefeller University) were maintained in Dulbecco's modified Eagle medium (Gibco, NY, USA) supplemented with 10% (vol/vol) fetal bovine serum, 50 IU/mL penicillin and streptomycin, in a humidified 5% (vol/vol) $CO_2$ incubator at 37°C.

### Multiple sequence alignment
The following HEV strains were used for sequence alignment: GenBank identifiers (IDs) or accession numbers M73218 (genotype 1 a, Burma strain), M74506 (2 a, Mex), JQ679013.1 (3,Kernow-C1/p6), AB197673 (4 a), AB573435 (5 a), AB856243 (6, wbJNN_13), KJ496144 (7, 180 C), and KX387866 (8, 48XJ). M73218 (genotype 1 a, Burma strain) was used as the reference strain for numbering. Sequence alignments were conducted with the SnapGene software (from Insightful Science; available at snapgene.com) using the multiple sequence alignment tool. HEV genotype alignments were conducted using CLUSTALW alignment algorithms embedded within the software.

Alignment of the HEV macro domain with the macrodomain of SINV (PDB: 4GUA) and CHIKV (PDB: 3GPG) were conducted with the SnapGene software (from Insightful Science; available at snapgene.com) using multiple sequence alignment tool with the local alignment Smith-Waterman algorithm.

## Hexa-cysteine motif bioinformatics

The HEV hexa-cysteine motif CxC[x]$_{11}$CC[X]$_8$CxC motif sequence identified was searched using Scan-Prosite (*de Castro et al., 2006*) on all UniProtKB/Swiss-Prot (release 2020_02 of 22-Apr-20: 562253 entries), UniProtKB/TrEMBL (release 2020_02 of 22-Apr-20: 0 entries) databases sequences using the regular expression [C-X-C-X(3,20)-C-C-X(3,20)-C-X-C]. The regular expression allows from 3 up to 20 residues in the two long stretches of amino acid residues where X can be any amino acid shown by previous bioinformatic analysis and sequence alignments across HEV genotypes (*Figure 1—figure supplement 1*). The search produced 429 hits in SwissProt and 32,764 hits in TrEMBL. We further refined the regular expression [C-X-C-X(11)-C-C-X(8)-C-X-C] to match the HEV hexa-cysteine motif exactly and produced 25 protein hits with no known function. We refined this search yet again to include the regular expression [C-X-C-X(10,12)-C-C-X(7,9)-C-X-C], which allows (±) one residue in each long stretch of amino acids where X can be any amino acid, and found 26 proteins hits with known functions (*Figure 3—figure supplement 1*) and were used to predict the function of the motif sequence.

## ORF1 protein structure predictions

FASTA files of each species of ORF1 (WT or mutants) were submitted to the AlphaFold algorithm (*Jumper et al., 2021*; DeepMind, United Kingdom, v. 2.0.0--model preset = monomer; or in the case of the HAV 3 C protease, v. 2.1.1--model preset = monomer) run on the Princeton Research Computing DELLA Cluster at Princeton University. Five models of each protein prediction were produced, and the best ranked model for each was used for subsequent analysis.

## Atomic distance calculations

Atomic distance calculations between the HEV ORF1 structure predications produced via AlphaFold (WT or point mutants) and known structures within ORF1 or distantly related viral macrodomains were done via the structure based alignment tool TM-Align (*Zhang and Skolnick, 2005*). Briefly, the known crystal structure of a region of the ORF1 pPCP/HVR (PDB: 6NU9 [*Proudfoot et al., 2019*]) was directly fed into the TM-Align software. The amphipathic 'thumb' of the RdRp solved via nuclear magnetic resonance (*Oechslin et al., 2022*), encountered a multi-mapping problem due to its alpha-helical nature and short amino acid sequence when fed directly into TM-Align; to generate the correct distance plot, the AlphaFold ORF1 prediction was trimmed of amino acids not corresponding to the region of the amphipathic RdRp thumb domain (Kc1/p6 AAs 1690–1708) utilizing UCSF Chimera. The trimmed ORF1 structure was aligned with the amphipathic thumb domain PDB, kindly provided by Jérôme Gouttenoire, using TM-Align. The distance plot generated between the AlphaFold ORF1 prediction and the macro domain of SINV was done by trimming a single chain of the trimer to remove amino acids outside the macrodomain of the SINV prediction P23pro-zbd (*Shin et al., 2012*; utilizing AAs 1342–1,509 of PDB: 4GUA) and aligning it to the ORF1 prediction using TM-Align. The CHIKV macrodomain PDB file 3GPG (*Malet et al., 2009*) was edited to remove the three additional chains comprising the hetero-tetramer and aligned with the ORF1 prediction.

## Molecular graphics and analysis

Molecular graphics and analyses performed with UCSF Chimera, developed by the Resource for Biocomputing, Visualization, and Informatics at the University of California, San Francisco, with support from NIH P41-GM103311 (*Pettersen et al., 2004*).

Note: Kernow c1/p6 strain contains an s17 insertion within the HVR, so amino acid positions shift downstream of this insertion at amino acids 751–806.

## Plasmid construction

To construct lentiviral constructs encoding ORF1 of Kernow C1/p6 (GenBank accession number JQ679013), the Kernow C1/p6 ORF1 cDNA was amplified by PCR from a plasmid encoding the FL infectious HEV clone Kernow C1/p6 (kindly provided by Suzanne Emerson, NIH) and then cloned into pLVX-IRES-zsGreen1 vector using an In-Fusion HD cloning kit (Clontech, Mountain View, CA, USA). The GAD mutant of ORF1 inactivating the polymerase was generated by QuikChange (Stratagene) site-directed mutagenesis. The HEV Kernow-C1 p6-Gluc (*Shukla et al., 2011*) and pSAR55-GLuc were kindly provided by Suzanne Emerson and Patricia Farci. pGEM-9zf-pSHEV3 and pGEM-7Zf(-)-TW6196E

encoding the infectious pSHEV3 (gt 3) and TW6196 (gt 4) clone, respectively were gifts from X.J. Meng. Site-directed mutagenesis of these plasmids for C483A or Pol(-) mutants was obtained with the QuikChange kit (Stratagene) using primers listed in *Supplementary file 1*. HEV Kernow-C1 p6-Gluc was used to generate the triplicate mutants for the alanine scanning mutagenesis of the entire pPCP domain, as well as the C434A, C457A, C459A, C471A, C472A, C481A, C483A, C563A, C483S, C483C (TGT), Y590A, Y590F, C483A-Y590A, C483G-Y590L, C483A (GCT), C483A (GCA), C483A (GCG), D248A, and H249A point mutants by QuikChange XL site-directed mutagenesis kit (Stratagene, La Jolla, CA, USA). All primers used for site-directed mutagenesis can be found in *Supplementary file 1*. AAAPG-HA tag-AAAPG insert was generated by amplifying out the HA tag sequence from pLVX ORF2-HA using primers listed in *Supplementary file 1*. pLVX Kernow C1/p6 ORF1 AAAPG-HA tag-AAAPG IRES zsGreen was generated via XhoI and XbaI digestion of pLVX IRES zsGreen plasmid and In-Fusion HD cloning of ORF1 AAAPG-HA tag-AAAPG from p6/BSR-2A-ZsGreen AAAPG-HA tag-AAAPG HVR using primers listed in *Supplementary file 1*. Overexpression constructs for ORF1 AAAPG-HA tag-AAAPG point mutants C483A, C563A, D248A, and H249A were generated from the parent pLVX Kernow C1/p6 ORF1AAAPG-Ha tag-AAAPG IRES zsGreen construct via QuikChange XL site-directed mutagenesis kit (Stratagene, La Jolla, CA, USA) using primers listed in *Supplementary file 1*.

## Generation of HEV reporter genomes

The generation of p6/BSR-2A-ZsGreen was described previously (*Nimgaonkar et al., 2021*). The generation of pSK-SAR55-Gluc, pGEM-9Zf-pSHEV3-Gluc, and pGEM-7Zf(-)-TW6196E-Gluc reporter constructs were described previously (*Ding et al., 2018b*). Generation of pSK-Kernow AAAPG-HA tag-AAAPG HVR GLuc was conducted via PCR linearization of pSK Kernow WT GLuc and In-Fusion HD cloning of AAAPG-HA tag-AAAPG cloned out of p6/BSR-2A-ZsGreen AAAPG-HA tag-AAAPG HVR using primers listed in *Supplementary file 1*.

All DNA fragments were cloned into the respective vectors using an In-Fusion HD cloning kit (Clontech, Mountain View, CA, USA). All constructs or primers used to construct the HEV reporter genomes in *Supplementary file 1* have been validated through Sanger sequencing and are available upon request.

## In vitro transcription assay and viral RNA transfection

HEV Kernow-C1 p6-Gluc, HEV Kernow-C1 p6 C483A-Gluc, HEV Kernow-C1 p6 GAD-Gluc, and all Kernow point mutant and alanine scanning triplite construct plasmids were linearized by MluI. pSAR55-Gluc, pSAR55 C483A-Gluc, and pSAR55 GAD-Gluc were linearized by BglII, pGEM-9Zf-pSHEV3-Gluc, pGEM-9Zf-pSHEV3 C483A-Gluc, and pGEM-9Zf-pSHEV3 GAD-Gluc were linearized by XbaI, and pGEM-7Zf(-)-TW6196E-Gluc, pGEM-7Zf(-)-TW6196E C483A-Gluc, and pGEM-7Zf(-)-TW6196E GAD-Gluc were linearized by SpeI. All capped viral RNA was in vitro transcribed from the corresponding linearized plasmid using the HiScribe T7 antireverse cap analog mRNA kit (New England Biolabs, Ipswich, MA, USA) according to the manufacturer's protocol. In vitro transcribed viral RNA was purified by LiCl precipitation following DNAse1 digestion. In vitro transcribed viral RNA was transfected into HUH7, HUH7.5, or HepG2C3A cells via the TransIT-mRNA transfection reagent (Mirus Bio LLC, Madison, WI, USA) according to the manufacturer's instructions.

## Gluc assays

Gluc activity was determined using Luc-Pair Renilla luciferase HS assay kit (GeneCopoeia, Rockville, MD, USA). Specifically, 10 μl of harvested cell culture medium was added per well of a 96-well solid white, flat-bottom polystyrene microplate (Corning, NY, USA), followed by the addition of Renilla luciferase assay substrate according to manufacturer protocol, and the detection of luminescence was performed using a Berthold luminometer (Bad Wildbach, Germany).

## Immunopurification of HA-tagged proteins

Immunopurification of HA tagged proteins for ICP-MS analysis was conducted using the Pierce MS-Compatible Magnetic IP Kit, protein A/G (Thermo Scientific, Waltham, MA, USA, Catalog number 90409). 750 μg of crude protein lysates of stably expressing ORF1 cells (WT or mutants) were subjected to each round of IP for subsequent analyses (ICP-MS or western blot). IP was done with rabbit anti-HA

antibody (Cell Signaling Technology, Danvers, MA, USA, catalog number C29F4) at a ratio of 1:50 in a laminar flow tissue culture hood that was sterilized with 70% ethanol and washed with MilliQ water prior to IP. The IP was conducted according to the manufacturer's instructions, and each collected fraction (crude cell lysate, unbound fraction, and IP eluate) was split into large (80% total volume) and small (20% total volume) fractions. The large fractions were used for downstream nitric acid/hydrogen peroxide digestion and ICP-MS analysis, whereas the small fraction was used for protein quantification and western blot analysis. This was necessary to reduce chances of contaminating divalent ions being introduced into the sample during non-ICP-MS characterization.

## HA-tagged immunopurified protein digestion

Eluted protein samples were freeze dried and placed at –80°C until nitric acid digestion. Briefly, protein samples were resuspended in 1.5 mL Nitric Acid 67–69% Optima, for Ultra Trace Elemental Analysis (Fisher Chemical, Fair Lawn, NJ, USA) in ICP-MS grade Teflon at 80°C for 24 hr in a laminar flow hood. 500 µL Hydrogen Peroxide (Optima, Fisher Chemical, Fair Lawn, NJ, USA) was added to each sample and incubated at 80°C for 24 hr in a laminar flow hood, when the samples were subsequently dried for 24 hr at 40°C. Fully digested samples were then resuspended in 1 mL 2% (vol/vol) nitric acid dilutes with MilliQ water, diluted in 1×, 2×, and 10× dilution series, and interrogated via ICP-MS (iCap, Thermo Scientific, Waltham, MA, USA).

## Inductively coupled plasma mass spectrometry

Samples were quantified via a single quadrupole iCap ICP-MS (Thermo Scientific, Waltham, MA, USA). Briefly, 1 mL of 2% (vol/vol) nitric acid running buffer blank was measured, followed by 1 mL per dilution of a dilution series (1×, 2×, 5×, 10×, and 50×) of the certified elemental standard 1643 F (National Institute of Standards and Technology, Gaithersburg, MD, USA) to generate an elemental calibration curve. All elements were measured in standard (STD) mode with the exception of lithium and iron, which were measured in kinetic mode to remove unwanted polyatomic interferences with the argon plasma. Samples were bracketed by an additional blank and STD curve to monitor instrument drift and ensure consistency throughout each experimental run. Samples were run via a series of three dilutions per sample, per run. Any measurement for a sample within the dilution series that fell outside of the dilution series range was excluded from further analysis. Contamination of trace elements from reagents and MilliQ water was monitored by processing MilliQ 'samples' through the entire immunopurification and protein digestion protocol on a run-by-run basis. Any trace metal analyses that were not statistically different from the MilliQ control 'sample' were excluded from further consideration. Iron and zinc ions were binned prior to statistical analysis for ease due to having with multiple isotopes.

## Immunofluorescence and confocal microscopy

Naïve HUH7 cells or HUH7 cells expressing ORF1 WT, WT-HA-tag, C483A-HA-tag, C563-HA-tag, D248A-HA-tag, or H249A-HA-tag were seeded onto separate glass coverslips (#1.5; 10 mm; Thomas Scientific, Swedesboro, NJ, USA) in a 24-well plate at 100,000 cells per well. 2 days post seeding, the cells were fixed with 4% PFA for 15 min and subsequently permeabilized in 0.25% Triton x-100 for 15 min. The rabbit anti-HA tag, C29F4 (Cell Signaling Technology, Danvers, MA, USA) primary antibody was used at a ratio of 1:1000 (V/V), and the AlexaFluor647 (goat anti-rabbit IgG [H+L], ThermoFisher Scientific, Waltham, MA, USA) secondary antibody was used at a final concentration of 1 µg/mL. All antibodies were diluted with PBS and incubated for 40 min at room temperature (RT). Hoechst 33342 (ThermoFisher Scientific, Waltham, MA, USA) was incubated at a final concentration of 1 µg/mL for 10 min at RT. The coverslips were then mounted onto glass microscopic slides (VWR International, Radnor, PA, USA) with 5 µL of ProLong gold antifade reagent (ThermoFischer Scientific, Waltham, MA, USA). The stained samples were imaged using the Nikon A1R-Si microscope (Nikon, Melville, NY, USA) in the Princeton University Confocal Microscopy Facility. The images were taken at 40× magnification. Images were then analyzed using Fiji (ImageJ2) image analysis software.

## Statistical analysis

All statistical analyses were performed using GraphPad Prism software version 9.3.1. One-way ANOVA with Dunnett's multiple comparison analysis or Brown-Forsythe one-way ANOVA with Dunnett's T3

multiple comparison analysis tests were used to test for statistical significance of the differences between the different group parameters in experiments utilizing the Gluc reporter replicon. p Values of less than 0.05 were considered statistically significant. All data sets were analyzed for and cleaned of outliers using the robust regression and outlier removal (ROUT) method.

## Materials availability statement

All materials generated by the Ploss lab will be available upon request from the corresponding author.

## Acknowledgements

We kindly thank Susan Emerson and Patricia Farci (NIAID) for providing us with the pSK SAR55, pBSK(+) Kc1 ORF1 WT GLuc plasmid, XJ Meng (Virginia-Maryland College of Veterinary Medicine) for providing us the plasmids pGEM-9zf-pSHEV3 and pGEM-7Zf(-)-TW6196E encoding the infectious pSHEV3 (gt 3) and TW6196 (gt 4) clone, respectively. HEV ORF1 RNA dependent RNA Polymerase structural data was kindly provided by Jérôme Gouttenoire. We would also like to thank Christina DeCoste and Katherine Rittenbach in the Molecular Biology flow cytometry core facility and Dr. Gary Laevsky and the Molecular Biology Confocal Microscopy Facility which is a Nikon Center of Excellence for their excellent technical support. We further thank Dr. Frederick Hughson for indispensable advice and expertise, as well as Jaden Shirkey and Kevin DAmico of the Hughson lab, and all members of the Ploss lab for critical discussions and comments throughout experimentation and preparation of the manuscript. Work in the lab is supported by grants from the National Institutes of Health (R01 AI138797, R01 AI107301, R01 AI146917, R01 AI153236 to AP), a Burroughs Wellcome Fund Award for Investigators in Pathogenesis (#101539 to AP) and funding from Princeton University. RL and SM were supported by the National Institute of General Medicine Sciences of the National Institutes of Health under Award Number T32GM007388. This material is based upon work supported by the National Science Foundation Graduate Research Fellowship under Grant No. (DGE-2039656) awarded to RL. The Molecular Biology Flow Cytometry Resource Facility is partially supported by the Cancer Institute of New Jersey Cancer Center Support grant (P30CA072720).

## Additional information

### Funding

| Funder | Grant reference number | Author |
| --- | --- | --- |
| National Institute of Allergy and Infectious Diseases | | Alexander Ploss |
| Burroughs Wellcome Fund | | Alexander Ploss |
| National Institute of General Medical Sciences | | Robert LeDesma Stephanie Maya |
| National Science Foundation | | Robert LeDesma |

The funders had no role in study design, data collection and interpretation, or the decision to submit the work for publication.

### Author contributions

Robert LeDesma, Conceptualization, Data curation, Formal analysis, Validation, Investigation, Visualization, Methodology, Writing - original draft, Writing - review and editing; Brigitte Heller, Investigation; Abhishek Biswas, Data curation, Software, Formal analysis, Investigation, Visualization, Methodology, Writing - review and editing; Stephanie Maya, Investigation, Methodology, Writing - review and editing; Stefania Gili, Formal analysis, Investigation, Methodology, Writing - review and editing; John Higgins, Resources, Supervision, Methodology, Writing - review and editing; Alexander Ploss, Conceptualization, Resources, Data curation, Formal analysis, Supervision, Funding acquisition, Validation, Investigation, Visualization, Methodology, Writing - original draft, Project administration

## Author ORCIDs

Robert LeDesma http://orcid.org/0000-0003-1435-4523
Alexander Ploss http://orcid.org/0000-0001-9322-7252

## Decision letter and Author response

Decision letter https://doi.org/10.7554/eLife.80529.sa1
Author response https://doi.org/10.7554/eLife.80529.sa2

## Additional files

### Supplementary files

• Supplementary file 1. Source data for inductively coupled plasma mass spectrometry (ICP-MS) experimental runs and primer sequences used in plasmid generation. Spreadsheet tabs 1–3: raw ion reads for inductively coupled plasma mass spectrometry data for each of the independent experimental runs, organized by date of run. Spreadsheet tab 4: primer sequences for each construct generated for use int his manuscript.

• MDAR checklist

• Source data 1. AlphaFold Predictions *Figures 4 and 5*. These files are the best ranked (ranked 0) predictions generated by AlphaFold of HEV ORF1, its associated point mutants, and the hepatitis A virus (HAV) 3 C protease.

### Data availability

All data generated or analyzed during this study are included in the manuscript and supporting file.

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
