## [Editor Report]

Your findings that polyprotein domains are likely to have exclusively structural functions is important to the field. It is often not appreciated that a large polyprotein is not merely a linear assembly of the final digestion products and must adopt particular conformations to support the ordered cleavages that occur.

---

## [Decision Letter]

**Decision letter after peer review:**

Thank you for submitting your article "Structural features stabilized by divalent cation coordination within hepatitis E virus ORF1 are critical for viral replication" for consideration by *eLife*. Your article has been reviewed by 2 peer reviewers, and the evaluation has been overseen by a Reviewing Editor and Arturo Casadevall as the Senior Editor. The following individual involved in review of your submission has agreed to reveal their identity: Nels C Elde (Reviewer #3).

The paper aims to provide structural and functional information on the hepatitis E virus replication complex. The study will be of interest to broad number of people studying at virus replication, since the replication complex are targets for therapeutic interventions.

Essential revisions:

1. The failure-to rescue-by-protease-alone experiment is not as convincing as the other arguments, because intramolcularly cleaving proteases are well-known in RNA viruses. It is plausible that cis-acting function of an actual protease within ORF1 is needed for its processing in order to complement a defective genome. The authors should cite the literature of intramolecular proteolysis (eg. Lindenbach) and modify their interpretation.

2. It would greatly strengthen the paper to directly rule out ORF1 processing with biochemical approaches like the purifications shown in Figure 6b. Are the additional bands on the western from the purification fragments of ORF1? If so, are these non-specific processing events? Can this be distinguished from a protease function of pPCP? If it isn't possible to gain more direct evidence, the final sentence of the discussion ruling out protease activity needs to be more measured.

3. The mass spectrometry data in Figure 6 shows about a ~20-40% decrease in zinc levels with the various mutants, which is about the same as some of the other ions used as a control. As a general observation, zinc binding proteins have an extremely tight binding affinity for zinc and the ion is usually necessary for proper folding. Similar experiments, looking at zinc ion levels in mutant proteins, show very low levels relative to wt (less than 10%). Currently, the data are not convincing, however it is appreciated that production of recombinant material for the study could be challenging. Perhaps producing a smaller fragment in bacteria or other eukaryotic systems would be more useful. At a minimum, better addressing the results of metal binding differences in Figure 6d and HEV biology would help bolster the manuscript, including controls where mutations don't alter metal binding to have better context for the results.

---

## [Author Response]

Essential revisions:1. The failure-to rescue-by-protease-alone experiment is not as convincing as the other arguments, because intramolcularly cleaving proteases are well-known in RNA viruses. It is plausible that cis-acting function of an actual protease within ORF1 is needed for its processing in order to complement a defective genome. The authors should cite the literature of intramolecular proteolysis (eg. Lindenbach) and modify their interpretation.

We thank this reviewer for their careful review and thoughtful comments. The reviewer brings up an interesting point; for other RNA viruses, such as HCV, *cis* activity of a protease requires multiple domains of the polyprotein (e.g. NS3/4A protease), and that not all singular protein domains are rescuable in *cis* or in *trans* (doi: 10.1371/journal.ppat.1004817) and may require a larger portion of the genome in order to exert their functions. Further, *cis-*acting activity is first required prior to function *in trans* in the case of several HCV non-structural proteins (doi: 10.1371/journal.ppat.1004817). To address this, we have altered the manuscript in the following ways (with additions/changes highlighted in yellow):

– The final sentence of the section outlining the trans-complementation results has been modified to say:

This trans-complementation platform provides further means to uncouple the putative functions of the pPCP or larger intramolecular regions of ORF1, e.g. polyprotein processing or modulation of the host cellular environment, from viral genome replication.

– The discussion where these results are expounded upon has been modified to include this possibility in the following way:

Of all the domains within ORF1, the functions of the putative PCP remain the most debated. Though evidence for and against proteolytic cleavage continues to mount on both sides, it is important to take the scientific results, as well as the functionality of the ORF1 protein, in context. Our data in this study has shown that the pPCP of ORF1 cannot function outside of the context of the full length protein, which is rather uncommon for many RNA viruses such as hepatitis A virus(Lemon et al., 1991), HCV(Yang et al., 2000), and flaviviruses such as Zika virus(Ding, Gaska, et al., 2018). While most characterized (+) ssRNA viruses rely on proteases to liberate individual gene products from their encoded polyprotein, HEV may be an exception. While it remains conceivable that host proteases may post-translationally process ORF1, there is rather limited evidence that subunits of ORF1 itself harbors proteolytic activity. Furthermore, if processing were to occur, it is likely that only a small fraction of ORF1 might be cleaved, as suggested previously (Metzger et al., 2022); however, the smaller species of ORF1 in the previously cited study were unable to be characterized by mass spectrometric analysis, leaving the processing of ORF1 still subject to debate. The inability for the putative HEV PCP to act outside of the context of the full length ORF1 protein suggests that it has some orthogonal activity, and that HEV ORF1 likely functions as one large multi-domain protein. However, other known viral proteases such as HCV’s NS3/4A possess *cis*-acting activity as well as *trans*-acting activity (Kazakov et al., 2015). With this in mind, the possibility remains that some undiscovered domain of ORF1 that possesses *cis*-acting processing activity is required to functionally rescue a defective genome in *trans*; results in favor if this possibility have yet to come to light.

2. It would greatly strengthen the paper to directly rule out ORF1 processing with biochemical approaches like the purifications shown in Figure 6b. Are the additional bands on the western from the purification fragments of ORF1? If so, are these non-specific processing events? Can this be distinguished from a protease function of pPCP? If it isn't possible to gain more direct evidence, the final sentence of the discussion ruling out protease activity needs to be more measured.

We thank this reviewer for their careful review and thoughtful comments and are excited to share previously acquired negative data on this subject.

Prior to the advent of the replication competent epitope tagged ORF1 (usable for ORF1 purification), we attempted MS/MS experiments with untagged ORF1 WT vs ORF1 C483A. We ran a 12% polyacrylamide gel with total protein lysates from stable expressing HepG2C3A cells , (using two different versions of replication incompetent WT ORF1 as comparisons (HA-ORF1-FLAG and ORF1 with a FLAG-tag in the HVR at a non-tolerable site)), and isolated novel bands between conditions (see Author response image 1 GelCode Blue stained gel, isolated bands highlighted by white rectangles [far left lane is naïve cell lysate so bands isolated differed from this lane]). We then worked with the Mass Spectrometry core at Princeton University to run an in-gel digest prior to tandem mass spectrometric analysis via a time-of-flight instrument.

Briefly, gel fragments were washed sequentially with Ammonium Bicarbonate (AB), AB/Acetonitrile mix, then Acetonitrile, and reduced and alkylated with TCEP and CAA prior to in gel digestion. Gels were digested with Trypsin, then quenched with formic acid and washed with formic acid/acetonitrile mixtures prior to lyophilization and reconstitution in formic acid for MS/MS analysis. MS/MS analysis was trained to observe ORF1 encoded sequences by feeding the algorithm FASTA sequences of ORF1 WT and ORF1 C483A, and all cleavage junctions of peptides were interrogated for novel non-trypsin cleavage junctions, though none were found. This early experiment in our interrogation of the potential processing of ORF1 combined with our reported results and the current literature (e.g. Metzger et al.) led us to the conclusion that ORF1 is not-processed at any appreciable level and acts as a large multi-domain protein. These experiments were cost-prohibitive and not repeated with the replication competent ORF1, also due to recent publications that produced similar results. (e.g. Metzger et al.)

**Author response image 1. sa2fig1:** 

3. The mass spectrometry data in Figure 6 shows about a ~20-40% decrease in zinc levels with the various mutants, which is about the same as some of the other ions used as a control. As a general observation, zinc binding proteins have an extremely tight binding affinity for zinc and the ion is usually necessary for proper folding. Similar experiments, looking at zinc ion levels in mutant proteins, show very low levels relative to wt (less than 10%). Currently, the data are not convincing, however it is appreciated that production of recombinant material for the study could be challenging. Perhaps producing a smaller fragment in bacteria or other eukaryotic systems would be more useful. At a minimum, better addressing the results of metal binding differences in Figure 6d and HEV biology would help bolster the manuscript, including controls where mutations don't alter metal binding to have better context for the results.

We thank this reviewer for their careful review and thoughtful comments.

We respectfully disagree with the reviewer’s suggestion of producing a smaller fragment in bacteria or separate eukaryotic system. Our results reported here aimed at dissecting the function of this region within the full context of ORF1, as results generated via expressing arbitrary domains of ORF 1 are difficult to interpret. It would be hard to argue that this region was natively folding outside of the intramolecular forces of the rest of the protein, as suggested by our AlphaFold data (Figure 5). However, we agree with the reviewer that the ICP-MS results could be bolstered with the inclusion of other replication competent and incompetent mutants that exist outside of the context of these predicted zinc-binding domains and subjected them to the same ICP-MS analysis. To this end, we generated epitope tagged versions of the replication deficient mutant that harbors a mutation in the RNA-dependent RNA polymerase (GDD mutated to GAD). This mutant is deficient likely due to an enzyme deficiency and not a folding deficiency (though this remains to be definitively proven). Further, we included a tolerable mutant found via the alanine scanning mutagenesis screen that was predicted to residue in an area of disorder that putatively serves as some sort of linker and thus not implicated in folding or ion binding (E489A; Figure 3).

Our results from these analyses (New Supplementary Figure 6, updated source data) suggest show that the Pol (-) mutant and the tolerable E489A mutant do not alter the ion binding potential of any ion in a statistically significant way, though the C483A still did. These new data combined with our initially submitted data suggest that mutating C483A inhibits the protein’s ability to bind zinc, and likely fold correctly for functionality.